# Alteration of circadian machinery in monocytes underlies chronic kidney disease-associated cardiac inflammation and fibrosis

Yuya Yoshida [1,6], Naoya Matsunaga [1,2,6], Takaharu Nakao[1], Kengo Hamamura [1], Hideaki Kondo[3], Tomomi Ide[4], Hiroyuki Tsutsui[4], Akito Tsuruta[1], Masayuki Kurogi[1], Michio Nakaya[5], Hitoshi Kurose[5], Satoru Koyanagi [1,2] & Shigehiro Ohdo [1,6✉]

Dysfunction of the circadian clock has been implicated in the pathogenesis of cardiovascular disease. The CLOCK protein is a core molecular component of the circadian oscillator, so that mice with a mutated *Clock* gene (*Clk/Clk*) exhibit abnormal rhythms in numerous physiological processes. However, here we report that chronic kidney disease (CKD)-induced cardiac inflammation and fibrosis are attenuated in *Clk/Clk* mice even though they have high blood pressure and increased serum angiotensin II levels. A search for the underlying cause of the attenuation of heart disorder in *Clk/Clk* mice with 5/6 nephrectomy (5/6Nx) led to identification of the monocytic expression of G protein-coupled receptor 68 (GPR68) as a risk factor of CKD-induced inflammation and fibrosis of heart. 5/6Nx induces the expression of GPR68 in circulating monocytes via altered CLOCK activation by increasing serum levels of retinol and its binding protein (RBP4). The high-GPR68-expressing monocytes have increased potential for producing inflammatory cytokines, and their cardiac infiltration under CKD conditions exacerbates inflammation and fibrosis of heart. Serum retinol and RBP4 levels in CKD patients are also sufficient to induce the expression of GPR68 in human monocytes. Our present study reveals an uncovered role of monocytic clock genes in CKD-induced heart failure.

[1] Department of Pharmaceutics, Faculty of Pharmaceutical Sciences, Kyushu University, Fukuoka, Japan. [2] Department of Glocal Healthcare Science, Faculty of Pharmaceutical Sciences, Kyushu University, Fukuoka, Japan. [3] Center for Sleep Medicine, Saiseikai Nagasaki Hospital, Katafuchi, Nagasaki, Japan. [4] Department of Cardiovascular Medicine, Faculty of Medical Sciences, Kyushu University, Fukuoka, Japan. [5] Department of Pharmacology and Toxicology, Facility of Pharmaceutical Sciences, Kyushu University, Fukuoka, Japan. [6] These authors contributed equally: Yuya Yoshida, Naoya Matsunaga, Shigehiro Ohdo. ✉email: ohdo@phar.kyushu-u.ac.jp

It is well known that chronic kidney disease (CKD) affects the functions of other organs[1,2], which causes complications such as cardiovascular disease and cranial nerve disorder[3,4]. Heart failure is the leading cardiovascular complication in CKD patients and its prevalence increases with declining renal function—such patients have an ~10- to 100-fold increased risk of cardiovascular mortality than patients without CKD[5,6]. In addition, more than 40% of patients with chronic heart failure also have CKD and this prevalence is directly correlated with the severity of CKD[7]. These clinical observations suggest a significant pathophysiological link between the heart and kidneys[8]. During the progression of CKD, hypertension is often caused by fluid volume expansion and activation of the renin–angiotensin–aldosterone system (RAAS)[9,10]. Although CKD-induced hypertension is closely associated with increased risks of cardiovascular morbidity and mortality, several humoral and cellular immune responses are also involved in the cardiovascular complications of CKD.

In mammals, behavior, physiology, and metabolism are subject to a well-controlled daily rhythm, generated by an internal self-sustained molecular oscillator referred to as the circadian clock[11]. As the expression of up to 10% of genes has been suggested to be under the control of the circadian clock[12], disruptions in the circadian clock system lead to the onset of numerous diseases. Indeed, several epidemiological analyses and laboratory animal studies reveal a relationship between circadian rhythms and cardiovascular function[13–15]. Therefore, impairment of the circadian clock may underlie altered cardiovascular homeostasis. However, our understanding of the role of clock genes in the CKD-induced cardiovascular dysfunction is limited.

The gene products of *Clock* and *Arntl* (also known as *Bmal1*) form a heterodimer that activates the transcription of *Period* (*Per*) and *Cryptochrome* (*Cry*) genes through E-box (5′-CACGTG-3′) enhancer elements. Once PER and CRY proteins reach a specific concentration, they attenuate CLOCK/ARNTL-mediated transactivation[16–18]. The alternating activation and suppression of the CLOCK/ARNTL-driven positive loop and PER/CRY-controlled negative loop result in a circadian oscillation in the molecular clock and also regulate 24 h variations in output physiology through the periodic activation/repression of clock-controlled genes[19,20]. *Clock* gene mutant (*Clk/Clk*) mice have a point mutation causing the deletion of exon 19 of the *Clock* gene, thus synthesizing mutant CLOCK protein deficient in transcriptional activity[21]. The mutant CLOCK protein causes low-amplitude rhythms in the expression of several genes, which leads to abnormal behavioral and physiological rhythms[16,21]. Many previous studies report the pathological states of *Clk/Clk* mice[15,22,23]. We and others investigated how the mutation of *Clock* gene affects the pathology of CKD[24,25]. In characterizing the hepatic metabolic status and inflammatory response of wild-type mice with 5/6 nephrectomy (5/6Nx), an experimental model of CKD characterized by the slow development of glomerulosclerosis, vascular sclerosis, tubulointerstitial fibrosis, and renal inflammation, we found that the abnormal increase in serum retinol levels in 5/6Nx wild-type mice induces the activation of caspase and apoptotic cell death in the kidney, further exacerbating the pathologies of CKD[26]. In contrast, 5/6Nx-induced renal inflammation and apoptotic cell death were attenuated in *Clk/Clk* mice[24], despite exhibiting high serum levels of retinol and retinol binding protein 4 (RBP4). Although abnormal retinoid homeostasis is often observed in patients with CKD[27], it remains unknown whether CKD-induced increase in retinol levels exacerbates cardiovascular disorders via alteration of *Clock* gene expression.

In this study, we demonstrate that cardiac inflammation and fibrosis are attenuated in 5/6Nx *Clk/Clk* mice even though they have a high blood pressure and increased levels of angiotensin II

and aldosterone. The infiltration of immune cells, especially monocytes, to the heart is also one of the leading causes of cardiac hypertrophy and fibrosis under cardiac load. Monocyte-derived macrophages orchestrate the inflammatory response by releasing cytokines and proteases, and activate myofibroblasts and immune cells[28–30]. An evaluation for the underlying mechanism of attenuation of cardiac inflammation and fibrosis in 5/6Nx mice leads to the identification of the monocytic expression of G protein-coupled receptor 68 (GPR68) as a key molecule exacerbating cardiac function during chronic renal failure. The 5/6Nx induces the expression of GPR68 in circulating monocytes by altering CLOCK and its heterodimer partner ARNTL (also known as BMAL1) whose transcriptional activity is increased by increasing serum levels of retinol and RBP4. Furthermore, serum levels of retinol and RBP4 in CKD patients are sufficient to induce the expression of GPR68 in human monocytes. Translation of these findings to humans will require clinical examination of the monocytic expression of the human *GPR68* gene, which we found to be functionally similar to mouse *Gpr68*.

## Results

**Mutation of *Clock* suppresses cardiac complications in 5/6Nx mice.** Subtotal nephrectomy, termed 5/6Nx, induces the progressive renal failure involving glomerulosclerosis and tubulointerstitial fibrosis[31,32]. In addition to these features of CKD, 5/6Nx wild-type mice exhibited high blood pressure ($P < 0.01$) and increased serum levels of angiotensin II, aldosterone ($P < 0.01$ for angiotensin II, $P < 0.01$ for aldosterone, respectively; Fig. 1a), creatinine, urea nitrogen ($P < 0.01$ for creatinine, $P < 0.01$ for urea nitrogen, respectively; Supplementary Fig. 1a), retinol, and RBP4 ($P < 0.01$ for retinol, $P < 0.01$ for RBP4, respectively; Supplementary Fig. 1b). As the serum levels of brain natriuretic peptide (BNP), a marker of cardiac dysfunction, were also increased in 5/6Nx wild-type mice ($P < 0.01$, Fig. 1a), we investigated the cardiac function and fibrosis status in this animal model. Increased end-diastolic interventricular septal thickness (Supplementary Fig. 2) and exacerbation of ventricular fibrosis ($P < 0.01$, Fig. 1b) were also detected in 5/6Nx wild-type mice. Inflammatory cytokines and fibrosis-related factors underlie CKD-induced heart failure[33,34]. The mRNA levels of inflammatory cytokines interleukin-6 (*IL-6*) and tumor necrosis factor-α (*Tnfα*) (Fig. 1c), and fibrosis-related factors α-smooth muscle actin (*αSma*) and pro-alpha2 chain of type I collagen (*Col1a2*) (Fig. 1d) were increased in the heart of 5/6Nx wild-type mice ($P < 0.01$). This suggests that the CKD model mice have cardiac complications similar to those observed in patients with chronic renal failure.

Contrary to wild-type mice, cardiac impairment and fibrosis were not observed in 5/6Nx *Clk/Clk* mice, although they had high blood pressure and increased serum levels of angiotensin II, aldosterone, retinol, and RBP4 (Fig. 1a–d and Supplementary Fig. 1b). Therefore, we hypothesized that the product of *Clock* gene is involved in the pathogenesis of CKD-induced heart failure. As CLOCK protein acts as a transcriptional factor for a variety of genes, we performed microarray analysis using RNA isolated from the cardiac ventricle of sham-operated (Sham) and 5/6Nx mice. By combining our present results and previous two transcriptome datasets (GSE54652 and GSE117488), we selected CLOCK-dependent 5/6Nx-induced genes by applying three criteria as follows: (1) the expression levels exhibited circadian oscillation in the mouse heart (Supplementary Table 1); (2) the presence of possible binding sites of CLOCK protein (Supplementary Table 2); and (3) expression levels in the heart of 5/6Nx wild-type mice were higher than those in Sham wild-type mice (Supplementary Table 3). Based on this analysis, *Cystin1* (*Cys1*), *EF-hand domain-containing protein D2* (*Efhd2*), *Gpr68*, and

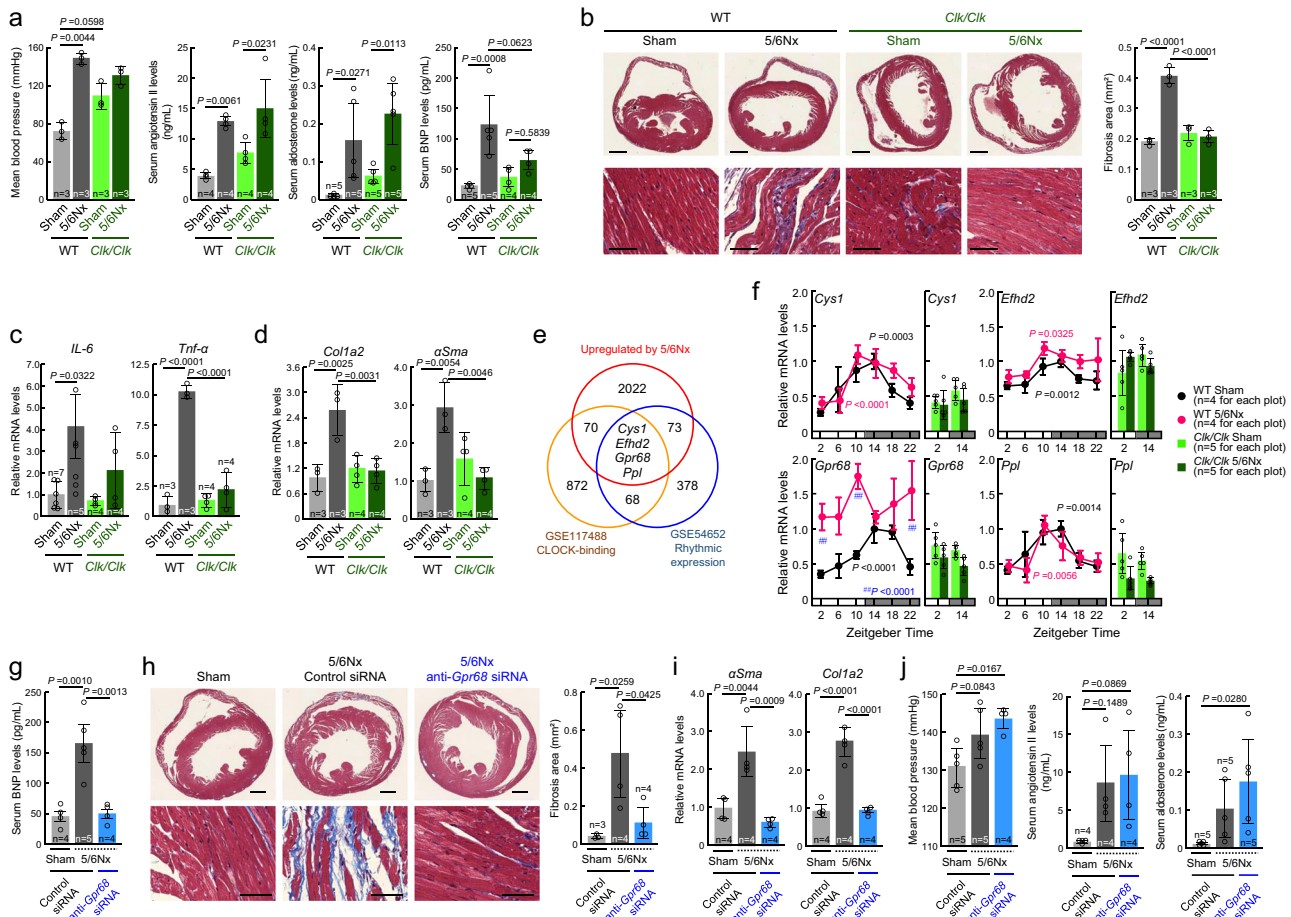

**Fig. 1 Mutation of *Clock* gene suppresses cardiac complications in 5/6Nx mice. a** Blood pressure and serum concentrations of angiotensin II, aldosterone, and BNP in wild-type (WT) and *Clk/Clk* mice with 5/6Nx. **b** Masson's trichrome staining of fibrosis (blue) in the cardiac ventricle of WT and *Clk/Clk* mice with 5/6Nx. The right panel shows the quantification of the fibrosis area under light microscopy. **c, d** The mRNA levels of *IL-6*, *Tnfα*, *Col1a2*, and *αSma* in the cardiac ventricle of WT and *Clk/Clk* mice with 5/6Nx. The mean value of the sham-operated wild-type group was set as 1.0. **e** Identification of CLOCK-dependent 5/6Nx-induced genes in the mouse heart. Three criteria were set to identify the genes as described in the "Results" and experimental procedure sections. **f** Temporal expression profiles of *Cys1*, *Efhd2*, *Gpr68*, and *Ppl* in the cardiac ventricle of WT and *Clk/Clk* mice with 5/6Nx. The mean peak value in the sham-operated wild-type group was set as 1.0. **g** Serum BNP concentrations in Sham and 5/6Nx mice after injection with control or anti-*Gpr68* siRNA. **h** Masson's trichrome staining in the cardiac ventricle of Sham and 5/6Nx mice after injection with control or anti-*Grp68* siRNA. The right panel shows the quantification of the fibrosis area under light microscopy. **i** The mRNA levels of *αSma* and *Col1a2* in the cardiac ventricle of Sham and 5/6Nx mice after injection with control or anti-*Grp68* siRNA. **j** Blood pressure levels and serum concentrations of angiotensin II and aldosterone in Sham and 5/6Nx mice after injection with control or anti-*Grp68* siRNA. For (**b** and **h**), scale bars indicate 1 mm (upper panel) and 50 μm (lower panel). For all panels, graphs show the mean ± SD of individual mice in independent experiments. Statistical significance was determined using two-way ANOVA (**a–d, f**), or one-way ANOVA (**g–j**) with Tukey–Kramer post hoc tests. Numbers and *P*-values are shown in each graphs. Source data are provided as a Source Data file.

*Periplakin* (*Ppl*) were screened as candidate CLOCK-dependent 5/6Nx-induced genes (Fig. 1e). The mRNA levels of these genes exhibited significant diurnal oscillation in the cardiac ventricle of Sham wild-type mice ($P < 0.01$, Fig. 1f) when they were maintained on a 12 h light–dark cycle (zeitgeber time (ZT); ZT0, lights on; ZT12, lights off). Of these, the levels of *Gpr68* mRNA in 5/6Nx wild-type mice increased at all examined time points, but such an increase in the *Gpr68* mRNA levels was not observed in *Clk/Clk* mice even after 5/6Nx (Fig. 1f). Therefore, we further focused on *Gpr68* as a CLOCK-dependent 5/6Nx-induced gene and investigated its function in CKD-induced heart failure.

**Downregulation of GPR68 alleviates CKD-induced cardiac inflammation.** To investigate whether GPR68 is involved in the exacerbation of cardiac dysfunction in 5/6Nx mice, the expression of GPR68 was downregulated by small interfering RNA (siRNA) using hemagglutinating virus of Japan envelope (HVJ-E). In this experiment, control or *Gpr68* siRNA was intravenously

injected into 5/6Nx wild-type mice once a week for 5 weeks (Supplementary Fig. 3a, b). Significant suppression of the 5/6Nx-induced increase in the serum BNP levels was observed in wild-type mice administered siRNA against *Gpr68* ($P < 0.01$, Fig. 1g). Downregulation of GPR68 expression also suppressed the 5/6Nx-induced cardiac fibrosis ($P < 0.01$, Fig. 1h), which was accompanied by repressed expression of inflammatory cytokines ($P < 0.01$, Supplementary Fig. 3c) and fibrosis-related factors ($P < 0.01$, Fig. 1i) in the ventricle of the heart. However, there was no improvement of the common features of CKD in GPR68-downregulated 5/6Nx wild-type mice, such as hypertension or increased serum levels of angiotensin II, aldosterone, or urea nitrogen (Fig. 1j and Supplementary Fig. 3d). These findings suggest that GPR68 is involved in the pathogenesis of CKD-induced cardiac inflammation and fibrosis.

**GPR68 expression in monocyte-derived cardiac macrophages.** To identify the types of cardiac cells expressing GPR68, we

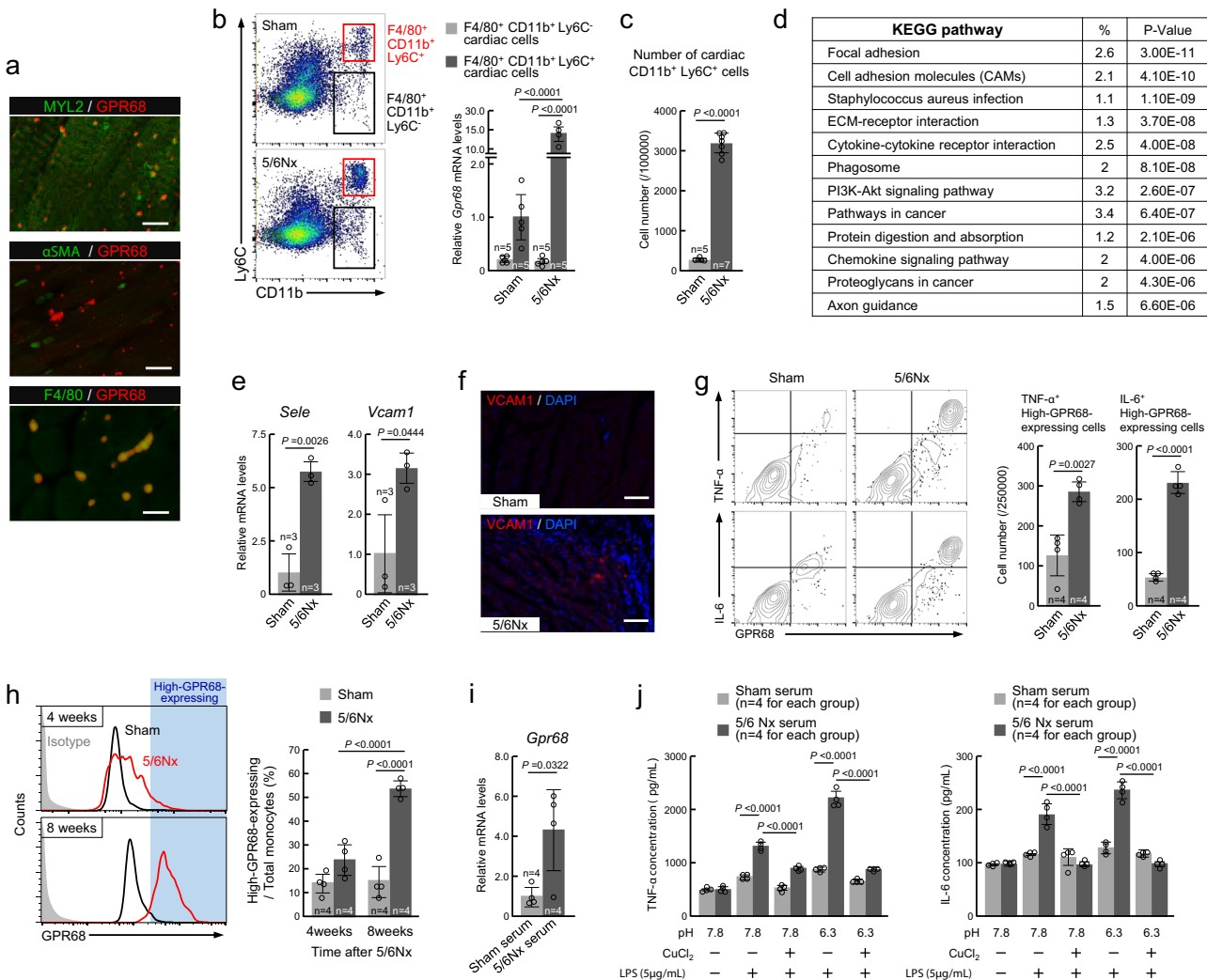

**Fig. 2 Inflammation response of high-GPR68-expressing monocytes in 5/6Nx mice. a** Double immunofluorescence labeling of GPR68 with MYL2, αSMA, and F4/80 in the cardiac ventricle of 5/6Nx mice. The scale bar indicates 20 μm. **b** The expression of *Gpr68* mRNA in cardiac macrophages. After separation of F4/80+/CD11b+ cells into Ly6C+ and Ly6C− populations, the mRNA levels of *Gpr68* in each population were measured by RT-PCR. **c** Increase in the number of cardiac-infiltrated CD11b+/Ly6C+ cells in 5/6Nx mice. **d** Functional analysis of the genes whose expression was changed in the cardiac ventricle of 5/6Nx mice using the KEGG database. **e** The mRNA levels of *Sele* and *Vcam1* in the cardiac ventricle of 5/6Nx mice. The mean value of the Sham group was set as 1.0. **f** Immunofluorescence labeling of VCAM1 (red) in the cardiac ventricle of Sham and 5/6Nx mice. The scale bar indicates 100 μm. **g** The number of TNFα- and IL-6-positive high-GPR68-expressing CD11b+/Ly6C+cells in the cardiac ventricle of Sham and 5/6Nx mice. **h** The ratio of high-GPR68-expressing monocytes in the blood of Sham and 5/6Nx mice at 4 or 8 weeks after nephrectomy. **i** The expression of *Gpr68* mRNA in primary cultured monocytes after incubation in the media containing 10% serum collected from Sham and 5/6Nx mice. **j** Release of TNFα and IL-6 from LPS-stimulated monocytes. Primary cultured monocytes were incubated in the media containing 10% serum collected from Sham and 5/6Nx mice for 24 h. Thereafter, cells were incubated in the presence or absence of 5 μg/mL of LPS and 0.2 mM CuCl₂ under pH 7.8 or 6.3 conditions for 8 h. For **i** and **j**, monocytes were isolated from naïve ICR mice and serum was collected from individual mice. For all panels, graphs show the mean ± SD of individual mice in independent experiments. Statistical significance was determined using two-way ANOVA with Tukey–Kramer post hoc tests (**b**, **h**, **j**) or two-tailed Student's *t*-tests (**c**, **e**, **g**, **i**). Numbers and *P*-values are shown in each graph. Source data are provided as a Source Data file.

performed double immunofluorescence labeling for GPR68 and cell type-specific markers: for cardiomyocytes, MYL2; for myoblasts, αSMA; and for macrophages, F4/80. In the ventricle in 5/6Nx wild-type mice, cells exhibiting GPR68 immunofluorescence were not double-labeled for MYL2 or αSMA (Fig. 2a), and most GPR68-positive cells were instead double-labeled with only F4/80 (Fig. 2a). These results are consistent with previous reports that *Gpr68* transcripts are enriched in immune cells[35].

Cardiac macrophages are classified into two main categories: F4/80+/CD11b+/Ly6C+ migrating macrophages originating from monocytes and F4/80+/CD11b+/Ly6C− resident macrophages[36]. Therefore, we used flow cytometry to identify more detailed cell types expressing GPR68 in the heart ventricle of 5/6Nx mice via indirect immunofluorescence using specific antibodies against F4/80, CD11b, and Ly6C (Supplementary Fig. 4). The mRNA levels of *Gpr68* in the F4/80+/CD11b+/Ly6C+ cell population were higher than those in F4/80+/CD11b+/Ly6C− cells (Fig. 2b), but the *Gpr68* mRNA levels were significantly increased in F4/80+/CD11b+/Ly6C+ cells prepared from the heart ventricle of 5/6Nx mice (*P* < 0.01), accompanied by an increased number of CD11b+/Ly6C+ cells (*P* < 0.01, Fig. 2c). The expression levels of adhesion molecules and chemokines involved in the regulation of monocyte infiltration were also increased in the heart ventricle of 5/6Nx mice (Fig. 2d–f and Supplementary Fig. 5). This suggests

that GPR68 in the heart ventricle of 5/6Nx mice is mainly expressed in infiltrated monocytes. The cardiac infiltration of monocytes may be promoted by adhesion molecules whose expression was induced by chronic renal failure.

**pH-dependent activation of high-GPR68-expressing monocytes to produce inflammatory cytokines.** A decrease in extracellular pH is often detected at the inflammatory site[37]. GPR68 is a proton-sensing receptor for detecting the extracellular acidic pH[38]. To examine whether high-GPR68-expressing monocytes function in inflammatory responses in the cardiac ventricle of 5/6Nx mice, we also conducted flow cytometry analysis to assess TNFα- or IL-6-positive (TNFα$^+$ or IL-6$^+$) high-GPR68-expressing monocytes in the cardiac ventricle of Sham and 5/6Nx wild-type mice. CD11b$^+$/Ly6C$^+$ cells were selected as monocyte-derived macrophages. Of these, TNFα$^+$ or IL-6$^+$ high-GPR68-expressing cells were counted via indirect immunofluorescence using their specific antibodies. The numbers of TNFα$^+$ or IL-6$^+$ high-GPR68-expressing monocytes were significantly increased in the cardiac ventricle of 5/6Nx mice ($P < 0.01$ for both, Fig. 2g). Compared with Sham mice, the number of high-GPR68-expressing monocytes was significantly increased in the blood of 5/6Nx wild-type mice ($P < 0.01$, Fig. 2h), suggesting that monocytic expression of GPR68 in 5/6Nx mice is induced by humoral factors in the blood. Indeed, GPR68 expression was also significantly induced in mouse primary monocytes when they were cultured in media supplemented with 10% serum prepared from 5/6Nx mice ($P < 0.01$, Fig. 2i).

To investigate the ability of high-GPR68-expressing primary monocytes to produce inflammatory cytokines, the cells were incubated in natural (pH 7.8) or acidic (pH 6.3) media in the presence of lipopolysaccharide (LPS). Under both natural and acidic conditions, the extracellular release of TNFα and IL-6 from high-GPR68-expressing primary monocytes was significantly increased by treatment with 5 μg/mL of LPS ($P < 0.01$ for both, Fig. 2j). The LPS-stimulated release of inflammatory cytokines was further increased when cells were incubated in acidic media. This pH-dependent activation of the inflammatory response of high-GPR68-expressing primary monocytes was suppressed by copper ions, an inhibitor of the proton-sensing function of GPR68 (Fig. 2j). Taken together, these findings suggest that the induction of GPR68 expression in monocytes is required for their activation in response to inflammatory stimuli, especially under acidic conditions.

**Infiltration of high-GPR68-expressing monocytes into the heart during chronic renal failure.** Monocytes are released from the bone marrow into the peripheral blood and they seed tissues throughout the body within a few days[39,40]. On the other hand, the majority of monocytes that infiltrate to inflammation sites in the heart, such as myocardial infarcts, originate from the spleen. Compared with Sham wild-type mice, the expression of GPR68 in monocytes was increased in the blood and spleen of 5/6Nx wild-type mice, but not in the bone marrow (Fig. 3a). Therefore, we prepared splenectomized 5/6Nx (Splx-5/6Nx) mice to investigate the role of the cardiac infiltration of monocytes in CKD-induced cardiac complication (Supplementary Fig. 6a). As expected, the Splx operation significantly suppressed 5/6Nx-induced cardiac infiltration of CD11b$^+$/Ly6C$^+$ cells ($P < 0.01$, Fig. 3b). This suppressive effect was unlikely due to the improvement of renal function, because the serum urea nitrogen levels were not significantly reduced in Splx-5/6Nx mice (Supplementary Fig. 6b). On the other hand, the 5/6Nx-induced increase in the serum BNP levels was significantly suppressed by splenectomy ($P < 0.01$, Fig. 3c). Indeed, no significant increase in mRNA levels of αSma, Col1a2, Tnfα, or IL-6 was detected in the heart of Splx-5/6Nx

mice (Fig. 3d). In addition, double immunofluorescence labeling experiments for GPR68 and F4/80 revealed that the number of double-labeled cells decreased in the ventricle of Splx-5/6Nx mice (Fig. 3e), consistent with the above results, and the number of CD11b$^+$/Ly6C$^+$ high-GPR68-expressing cells was significantly lower in the ventricle of Splx-5/6Nx mice ($P < 0.01$, Fig. 3f) even though levels of adhesion molecules were upregulated in the heart (Supplementary Fig. 6c, d). These findings suggest that high-GPR68-expressing monocytes infiltrate from the spleen into the heart of 5/6Nx mice; thus, the cardiac infiltration of monocytes may be closely related to the CKD-induced inflammation and fibrosis in heart.

**Downregulation of GPR68 in monocytes alleviates CKD-induced cardiac inflammation.** To investigate whether high-GPR68-expressing monocytes are involved in the exacerbation of cardiac dysfunction in mice with chronic renal failure, monocytes were collected from the spleen of healthy mice and cells were infected with lentivirus-expressing microRNA (miRNA) against Gpr68. Control or GPR68-downregulated monocytes were injected intravenously into Splx-5/6Nx mice every 3 days for 2 weeks (Fig. 4a). Green fluorescence from the lentiviral green fluorescent protein (GFP) reporter was detected in the heart ventricle of Splx-5/6Nx mice injected with both control and anti-Gpr68 miRNA-transduced monocytes (Fig. 4b upper), but infiltration of high-GPR68-expressing cells was noted in the heart ventricle of Splx-5/6Nx mice injected with control miRNA-transduced monocytes (Fig. 4b lower), suggesting that Splx-5/6Nx mice still have the potential to induce GPR68 expression in allogeneic transplanted monocytes. The high-GPR68-expressing cells were merged with GFP-positive cells, confirming the infiltration of allogeneic transplanted monocytes into the heart of Splx-5/6Nx mice. As observed in 5/6Nx mice, the expression of GPR68 in transplanted monocytes of Splx-5/6Nx mice was also induced by humoral factors in the blood whose levels likely increase during chronic renal failure. On the other hand, cardiac infiltration of GPR68-expressing cells was undetectable in Splx-5/6Nx mice injected with anti-Gpr68 miRNA-transduced monocytes, although GFP fluorescence from lentivirus-transfected monocytes was observed in the ventricle of the heart (Fig. 4b right).

The allogeneic transplantation of control miRNA-transduced monocytes increased serum BNP levels ($P < 0.01$, Fig. 4c) and cardiac fibrosis ($P < 0.05$, Fig. 4d) in Splx-5/6Nx mice, but no significant BNP increase or cardiac fibrosis was detected in Splx-5/6Nx mice transplanted with GPR68-downregulated monocytes. Similarly, the mRNA expression of inflammatory cytokines (Fig. 4e) and fibrosis-related factors (Fig. 4f) in the heart ventricle of Splx-5/6Nx mice was increased by allogeneic transplantation of control miRNA-transduced monocytes, whereas it was not increased in Splx-5/6Nx mice transplanted with GPR68-downregulated monocytes. In contrast, increased serum urea nitrogen levels in Splx-5/6Nx mice were not improved by the transplantation of GPR68-downregulated monocytes (Fig. 4g). Although intravenous injection of GPR68-expressing plasmids using HEV-J into healthy mice induced the expression of GPR68 in circulating monocytes, the number of GPR68-expressing monocytes was not significantly increased in the heart ventricle, probably due to the limited expression of adhesion molecules (Supplementary Fig. 7a–d). The increased GPR68 expression in the circulating monocytes in healthy mice was also unable to increase serum BNP levels or heart fibrosis (Supplementary Fig. 7e, f). These findings suggest that cardiac infiltration of high-GPR68-expressing monocytes play an important role in the exacerbation of CKD-induced inflammation and fibrosis in the heart.

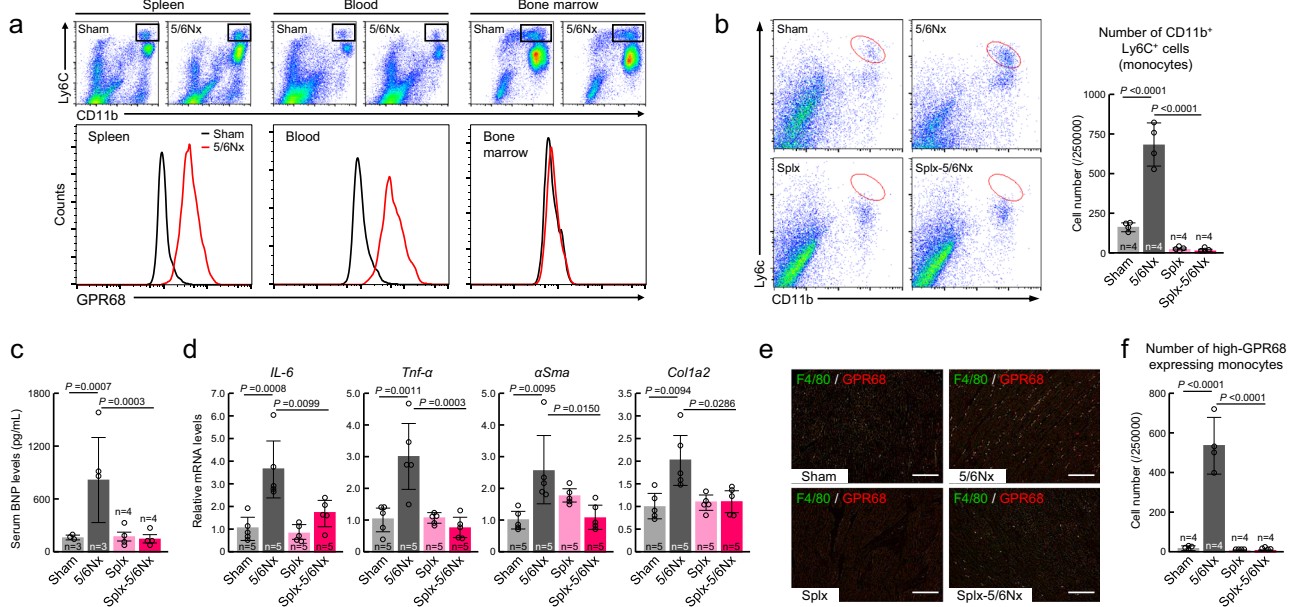

**Fig. 3 Splenectomy reduces cardiac infiltration of monocytes and heart failure in 5/6Nx mice. a** Induction of GPR68 expression in circulating monocytes in 5/6Nx mice. Flow cytometry analysis was performed to detect high-GPR68-expressing CD11b$^+$/Ly6C$^+$ cells in the spleen, blood, and bone marrow of Sham and 5/6Nx mice. **b** Decrease in the number of cardiac-infiltrated CD11b$^+$/Ly6C$^+$ cells in splenectomized (Splx)-5/6Nx mice. Left panel shows the flow cytometry histogram of CD11b$^+$/Ly6C$^+$ cells isolated from the cardiac ventricle of Sham, 5/6Nx, Splx, and Splx-5/6Nx mice. The number of CD11b$^+$/Ly6C$^+$ cells, indicated by red circles, was quantified and the data are shown in the right panel. **c** Serum concentrations of BNP in Sham, 5/6Nx, Splx, and Splx-5/6Nx mice. **d** The mRNA levels of inflammatory cytokines (*Tnf-α* and *IL-6*) and fibrosis-related factors (*αSma* and *Col1a2*) in the cardiac ventricle of Sham, 5/6Nx, Splx, and Splx-5/6Nx mice. The mean value of the sham-operated wild-type group was set as 1.0. **e** Double immunofluorescence labeling of GPR68 with F4/80 in the ventricle slices prepared from Sham, 5/6Nx, Splx, and Splx-5/6Nx mice. High-GPR68-expressing cells are double-labeled (yellow) with F4/80. The scale bar indicates 50 μm. **f** Decrease in the number of cardiac-infiltrated high-GPR68-expressing monocytes in Splx-5/6Nx mice. For all panels, graphs show the mean ± SD of individual mice in independent experiments. Statistical significance was determined using two-way ANOVA with Tukey–Kramer post hoc tests (**b–d**, **f**). Numbers and *P*-values are shown in each graph. Source data are provided as a Source Data file.

**Disruption of the circadian rhythm of clock gene expression in 5/6Nx monocytes upregulates GPR68 expression**. As *Gpr68* was identified as a CLOCK-dependent 5/6Nx-induced gene (Fig. 1f), we investigated the relationship between CLOCK protein and CKD-induced expression of GPR68 in monocytes. The mRNA levels of typical circadian genes, *Clock*, *Arntl*, *Per1*, and *Nr1d1*, exhibited significant diurnal oscillation in circulating monocytes of Sham wild-type mice at 8 weeks after the operation (Fig. 5a). In contrast, the mRNA levels of these circadian genes were increased in 5/6Nx wild-type mice. Among these, nuclear protein levels of CLOCK and ARNTL were also increased in 5/6Nx wild-type mice (Fig. 5b). As observed in the heart ventricle of 5/6Nx wild-type mice (Fig. 1f), oscillation of the monocytic expression of *Gpr68* was disrupted (Fig. 5c). The mRNA levels of *Gpr68* in monocytes of 5/6Nx wild-type mice continued to increase throughout the day.

CLOCK protein forms heterodimers with ARNTL, which regulate the expression of target genes through an E-box (CANGTG) element[16]. Sequence analysis using an online genome browser (JASPAR[41]; http://jaspar.genereg.net) revealed three E-box motifs located upstream of the *Gpr68* gene in mice (Fig. 5d) and all mammals examined, including rats, rhesus monkeys, and humans (Supplementary Fig. 8). Chromatin immunoprecipitation (ChIP) experiments also suggested that the binding of CLOCK and ARNTL to the E-box elements in the upstream region of the mouse *Gpr68* was time-dependently altered in the monocytes of Sham wild-type mice (*P* < 0.01, Fig. 5e). The amount of CLOCK and ARNTL binding to the E-box elements was increased in 5/6Nx wild-type mice during the day and at night (*P* < 0.01).

To further investigate the role of CLOCK in the regulation of 5/6Nx-induced expression of *Gpr68*, primary monocytes were

prepared from wild-type or *Clk/Clk* mice. As shown in Fig. 2i, the expression levels of *Gpr68* mRNA were significantly increased in naive wild-type monocytes when cells were incubated in media supplemented with 10% serum collected from 5/6Nx wild-type mice (*P* < 0.01, Fig. 5f). In contrast, such serum-induced expression of *Gpr68* was not observed in monocytes prepared from *Clk/Clk* mice (Fig. 5f). Taken together, CLOCK may mediate the 5/6Nx-induced expression of GPR68 in monocytes, which is caused by humoral factors in the blood.

**Serum retinol accumulation during chronic renal failure induces GPR68 expression in mouse monocytes**. To identify the serum factor that induces the expression of *Gpr68* in monocytes, we conducted one additional microarray experiment using RNA collected from monocytes of Sham and 5/6Nx mice at 8 weeks after nephrectomy (Supplementary Table 4). Based on functional analysis of genes using the Kyoto Encyclopedia of Genes and Genomes (KEGG) database[42], four biological pathways of "Environmental Information Processing" were significantly enriched (Fig. 6a). Of these, the Janus kinase-signal transducer and activator of transcription (JAK/STAT) signaling pathway is often activated by inflammatory cytokines. As observed in our previous study[26], serum levels of TNFα and IL-6 were increased in 5/6Nx mice at 8 weeks after nephrectomy (Supplementary Fig. 9). Serum concentrations of TNFα and IL-6 in 5/6Nx mice reached a maximum of 88 and 47 ng/mL, respectively. However, treatment of mouse primary cultured monocytes, with 1000 ng/ml of TNFα and 100 ng/mL of IL-6, had negligible effects on the mRNA expression of *Gpr68* (Fig. 6b). In addition to inflammatory cytokines, angiotensin II and retinol/RBP, whose serum

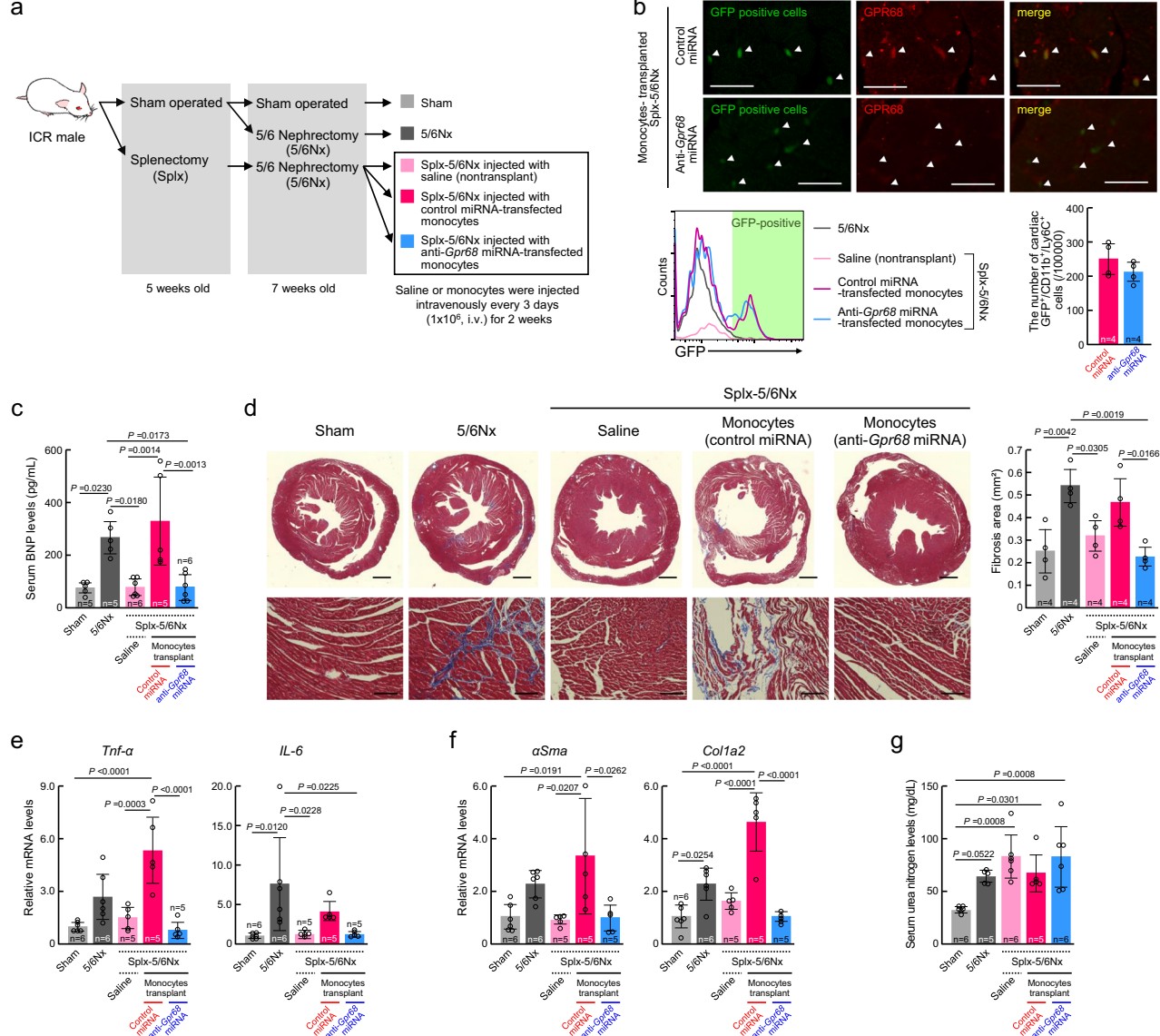

**Fig. 4 Role of high-GPR68-expressing monocytes in CKD-induced cardiac inflammation and fibrosis. a** Schematic experimental procedure for allogeneic transplantation of monocytes into Splx-5/6Nx mice. Monocytes infected with lentivirus-expressing control or anti-*Gpr68* miRNA were intravenously injected into Splx-5/6Nx mice every 3 days for 2 weeks after the 5/6 nephrectomy operation. **b** Upper panels show immunofluorescence labeling of GPR68 (red) in the cardiac ventricle slices prepared from Splx-5/6Nx mice at 2 weeks after initiating the injection of monocytes infected with lentivirus-expressing control or anti-*Gpr68* miRNA. Green fluorescence is derived from the lentiviral GFP reporter. The scale bar indicates 20 μm. Lower panels show the flow cytometry histogram of GFP fluorescence of $CD11b^+/Ly6C^+$ cells in the heart. GFP-positive cells were detected only in monocyte transplanted mice. **c** Serum BNP concentrations in Splx-5/6Nx mice at 2 weeks after initiating the injection of control or GPR68-downregulated monocytes. **d** Downregulation of GPR68 in monocytes ameliorated CKD-induced cardiac fibrosis. The left panel shows Masson's trichrome staining of tissue fibrosis in blue. Scale bars indicate 1 mm (upper panel) and 50 μm (lower panel). The right panel shows the quantification of the fibrosis area under light microscopy. **e**, **f** The mRNA levels of inflammatory cytokines (*Tnf-α* and *IL-6*) and fibrosis-related factors (*αSma* and *Col1a2*) in the cardiac ventricle of Splx-5/6Nx mice at 2 weeks after initiating the injection of control or GPR68-downregulated monocytes. **g** Serum urea nitrogen levels in Splx-5/6Nx mice at 2 weeks after initiating the injection of control or GPR68-downregulated monocytes. For all panels, graphs show the mean ± SD of individual mice in independent experiments. Statistical significance was determined using one-way ANOVA with Tukey–Kramer post hoc tests (**c**–**g**). Numbers and *P*-values are shown in each graph. Source data are provided as a Source Data file.

concentrations are often increased in CKD patients, activate the JAK/STAT signaling pathway[43,44]. Serum levels of angiotensin II, retinol, and RBP4 in 5/6Nx mice were increased to a maximum 15 ng/mL, 2.8 μM, and 4.9 μM, respectively (Fig. 1a and Supplementary Fig. 1). Although no significant induction of *Gpr68* expression was detected in mouse primary monocytes after treatment with 100 ng/mL of angiotensin II (Fig. 6c), the mRNA levels of *Gpr68* in mouse primary monocytes were dose-

dependently increased by retinol and RBP4 (*P* < 0.01, Fig. 6c). Significant *Gpr68* mRNA expression was detected when cells were treated with >2 μM retinol and RBP4 (*P* < 0.05), and 3 μM retinol and RBP4 were sufficient to increase the protein levels of GPR68 (*P* < 0.01, Fig. 6d). In addition, treatment of mouse primary monocytes with 3 μM retinol and RBP4 also increased the phosphorylation of nuclear STAT5 (phos-STAT5) protein (*P* < 0.01, Fig. 6e), and increased the mRNA levels of *Clock* and *Arntl*

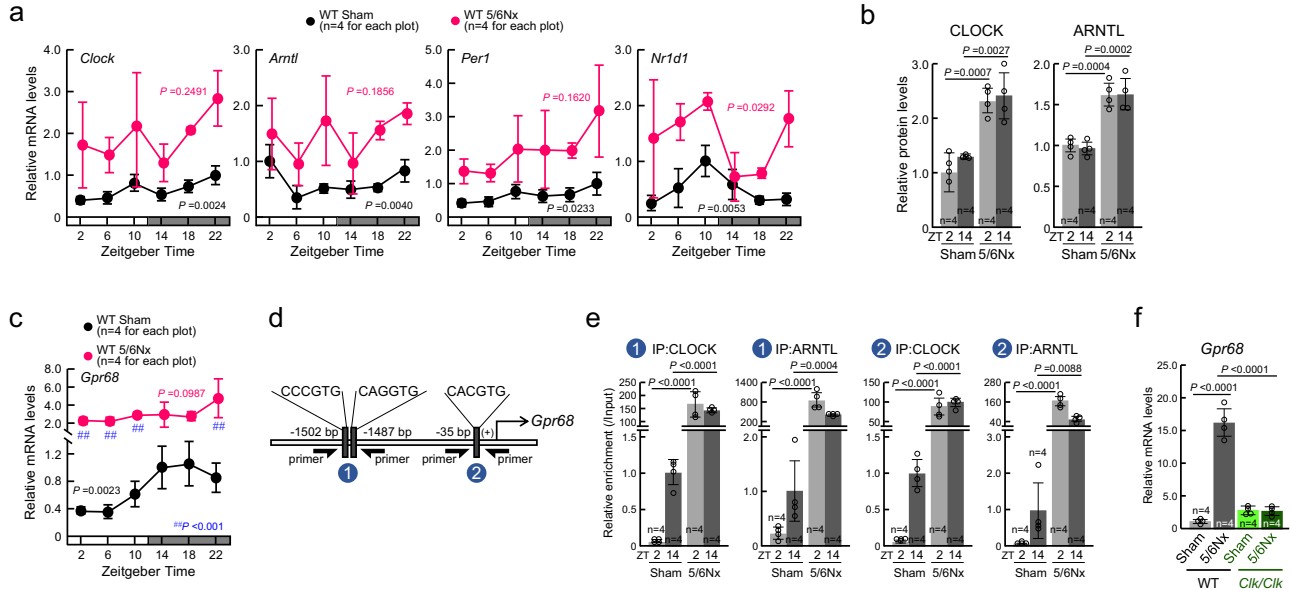

**Fig. 5 Disruption of clock gene expression in 5/6Nx monocytes upregulates GPR68 expression. a** Temporal mRNA expression profiles of *Clock*, *Arntl*, *Per1*, and *Nr1d1* in the circulating monocytes of Sham and 5/6Nx mice. **b** Temporal expression profiles of CLOCK and ARNTL proteins in the monocytes of Sham and 5/6Nx mice. **c** Temporal mRNA expression profiles of *Gpr68* in the monocytes of Sham and 5/6Nx mice. **d** Schematic representation of the mouse *Gpr68* gene. The numbers indicate the distance from the transcription start site (+1). Black rectangles, E-box. The circled numbers (blue circles) indicate the location on the gene where each of the different primer sets localize for analysis of chromatin immunoprecipitation (ChIP). **e** The binding of endogenous CLOCK and ARNTL to the *Gpr68* upstream region in the monocytes of Sham and 5/6Nx mice at ZT2 and ZT14. **f** The expression of *Gpr68* mRNA in primary cultured monocytes, which were isolated from naive wild-type (WT) or *Clk/Clk* mice. The mRNA levels of *Gpr68* were assessed after treatment with serum from Sham or 5/6Nx WT mice for 24 h. For (**a–c**) and **e**, the mean peak value of the Sham group was set as 1.0. For (**a** and **c**), there was a significant time-dependent variation in the mRNA levels of *Clock*, *Arntl*, *Per1*, *Nr1d1*, and *Gpr68* in the Sham group (one-way ANOVA; P-value are shown in each graphs). For all panels, graphs show the mean ± SD of individual mice in independent experiments (n = 4 for all groups). Statistical significance was determined using two-way ANOVA with Tukey–Kramer post hoc tests (**b**, **e**, **f**). Numbers and P-values are shown in each graph. Source data are provided as a Source Data file.

($P < 0.01$, Fig. 6f). The JASPAR analysis for up- and downstream regions from the transcriptional start site revealed no possible STAT5-binding site in the mouse *Gpr68* gene, but several STAT5-binding sites were located in the upstream regions of the *Clock* and *Arntl* genes (Fig. 6g). The ChIP experiment revealed that the amount of STAT5 binding to the upstream region of *Clock* and *Arntl* genes in mouse primary monocytes was increased by treatment with 3 μM of retinol and RBP4 ($P < 0.01$, Fig. 6h). A similar increase in STAT5 binding in the upstream region of *Clock* and *Arntl* genes was observed in monocytes collected from 5/6Nx mice at 8 weeks after nephrectomy ($P < 0.01$, Fig. 6i). In addition, treatment of mouse primary monocytes with 3 μM retinol and RBP4 increased CLOCK/ARNTL binding to the upstream region of *Gpr68* ($P < 0.01$, Fig. 6j), suggesting that retinol-induced expression of GPR68 is caused by CLOCK/ARNTL-mediated transactivation. Stimulation of STRA6 by retinol-bound RBP4 results in the activation of STAT5[43]. Downregulation of STRA6 in mouse primary monocytes by siRNA failed to induce the phosphorylation of STAT5 and the expression of CLOCK, ARNTL, and GPR68, even when the cells were incubated in the media supplemented with 10% serum collected from 5/6Nx mice (Fig. 6k). After intracellular permeation, retinol is converted into retinoic acid (Supplementary Fig. 10a), which activates retinoic acid receptor (RAR) and retinoid X receptor (RXR). These receptors form heterodimers and induce the expression of their target genes. Retinol-free (unbound) RBP4 also stimulates Toll-like receptor 4 and activates mitogen-activated protein kinases (Supplementary Fig. 10a). The increase in mRNA levels of *Gpr68*, *Clock*, and *Arntl*, in addition to increased phosphorylation of STAT5, were not observed in the cultured monocytes treated with retinoic acid or RBP4 alone (Supplementary Fig. 10b). Furthermore, treatment of monocytes with the RAR inhibitor Ro41-5253 had a negligible effect on the expression of *Gpr68*, *Clock*, and *Arntl*, and phosphorylation state of STAT5 (Supplementary Fig. 10b). This suggests that STRA6 is essential for retinol-induced expression of GPR68 in monocytes.

Vitamin A is derived from the diet, mostly as retinol esters. As retinol is the main circulating retinoid, dietary deficiency of vitamin A reduces serum retinol levels[45–47]. Feeding of 5/6Nx mice with a vitamin A-free diet (Supplementary Fig. 11a) reduced the serum levels of both retinol and RBP4 (Fig. 7a), accompanying the suppressed mRNA expression of *Gpr68*, *Clock*, and *Arntl* in circulating monocytes (Fig. 7b). Although serum urea nitrogen levels and cardiac expression of the adhesion molecule VCAM1 were not reduced in 5/6Nx mice fed the vitamin A-free diet (Supplementary Fig. 11b–d), the 5/6Nx-induced cardiac infiltration of GPR68-expressing monocytes (Fig. 7c) increase in serum BNP levels (Fig. 7d) and fibrosis in the heart (Fig. 7e) were suppressed by dietary deficiency of vitamin A. On the other hand, intraperitoneal administration of retinol to healthy mice induced GPR68 expression in the circulating monocytes, but retinol-treated healthy mice did not exhibit cardiac infiltration of monocytes, VCAM1 expression, or characteristics of heart failure (Supplementary Fig. 12). This suggests that the CKD-induced increases in serum retinol and RBP4 induce the monocytic expression of GPR68 and cardiac infiltration of GPR68-expressing cells may exacerbate CKD-induced inflammation and fibrosis in heart.

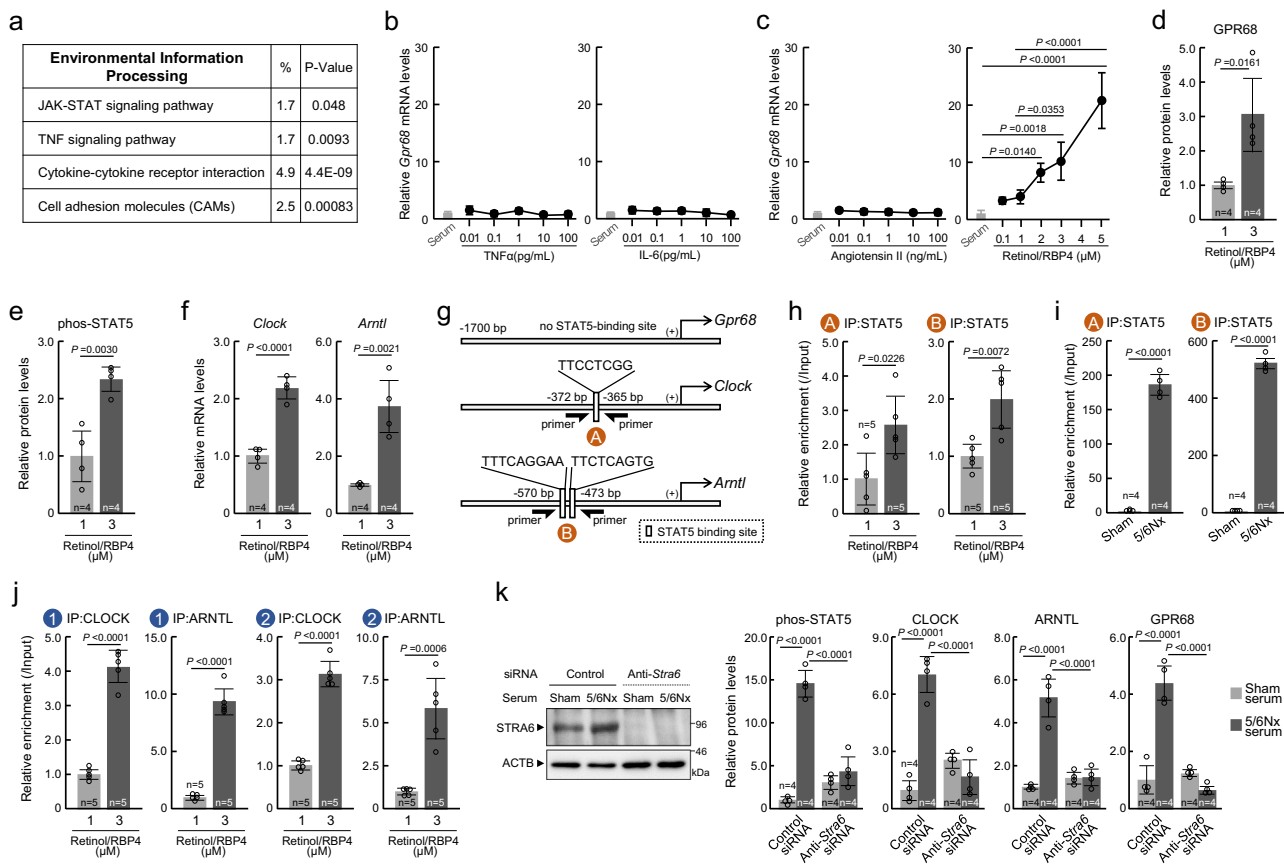

**Fig. 6 Increased serum retinol levels induce GPR68 expression in monocytes. a** Functional analysis of the genes using the KEGG database. Detailed data are presented in Supplementary Table 5. **b**, **c** The mRNA levels of *Gpr68* in mouse primary cultured monocytes after treatment with TNFα, IL-6, angiotensin II, or retinol/RBP4 at the indicated concentrations for 24 h. **d**–**f** The protein levels of GPR68 and phosphorylated STAT5 (phos-STAT5), and the mRNA levels of *Clock* and *Arntl* in mouse primary cultured monocytes after treatment with 1 or 3 μM of retinol/RBP4 for 24 h. **g** Schematic representation of the upstream region of the mouse *Gpr68*, *Clock*, and *Arntl* genes. The numbers indicate the distance from the transcription start site (+1). The circled letters (orange circles) indicate the location on the gene where each of the different primer sets localize for analysis of ChIP. **h**, **i** ChIP analysis of retinol/RBP4-induced changes in the binding amount of STAT5 protein to the upstream region of *Clock* and *Arntl* genes in mouse primary cultured monocytes or the collected monocytes prepared from Sham and 5/6Nx mice. The primer sets used qPCR are illustrated in **g**. **j** ChIP analysis of retinol/RBP4-induced changes in the binding amount of CLOCK and ARNTL protein to the upstream region of *Gpr68* genes in mouse primary cultured monocytes. Using primer sets illustrated in Fig. 5d. **k** Left panel shows the protein levels of STRA6 in monocytes transfected with siRNA against STRA6 (uncropped images are presented in Supplementary Fig. 21). Right panels show STRA6-dependent phosphorylation of STAT5 and expression of GPR68, CLOCK, and ARNTL by serum from Sham and 5/6Nx mice in mouse primary cultured monocytes. For all panels, graphs show the mean ± SD of individual mice in independent experiments (**h**, **j** $n = 5$; the others $n = 4$ for each group). Statistical significance was determined using one-way ANOVA with Dunnett's post hoc tests (**c**), two-tailed Student's *t*-tests (**d**–**f**, **h**–**j**), or two-way ANOVA with Tukey–Kramer post hoc tests (**k**). Numbers and *P*-values are shown in the graph. Source data are provided as a Source Data file.

**Retinol-induced expression of GPR68 in human monocytes.** The DNA sequences of STAT5-binding sites in the mouse *Clock* and *Arntl* genes were conserved in the human *CLOCK* and *ARNTL* genes (Supplementary Fig. 13). We also found multiple E-box elements in the upstream region of the human *GPR68* gene (Supplementary Fig. 8). Thus, we investigated whether retinol induces the expression of GPR68 in human monocytes. Similar dose-dependent induction of *GPR68* mRNA expression by retinol and RBP4 was detected in the human primary monocytes (Fig. 8a). Significant *GPR68* expression was observed when cells were treated with >2 μM retinol and RBP4 ($P < 0.01$). Furthermore, treatment of human primary monocytes with 3 μM retinol and RBP4 increased the phosphorylation of STAT5 protein ($P < 0.01$, Fig. 8b) and the binding of STAT5 to the upstream region of the human *CLOCK* and *ARNTL* genes ($P < 0.01$, Fig. 8c, d). Similarly, binding of CLOCK and ARNTL to the upstream region of the human *GPR68* gene was increased in monocytes treated with retinol and RBP4 ($P < 0.01$, Fig. 8d). The treatment of

human primary monocytes with retinol and RBP4 also increased the extracellular release of TNFα and IL-6 after stimulation with 5 μg/mL of LPS, but the increased release of inflammatory cytokines was suppressed by copper ions (Fig. 8e), confirming that the retinol-RBP4-induced expression of GPR68 in human monocytes activates their inflammatory response.

Based on these observations, we further investigated whether the serum accumulation of retinol and RBP4 in CKD patients induces *GPR68* expression in human primary monocytes. As serum RBP4 levels are also altered in patients with diabetes[48], we collected serum from CKD patients without diabetes. There were no abnormal total protein or total cholesterol levels, or sugar concentrations in collected serum samples (Supplementary Fig. 14a). In addition, RBP4 levels in the serum of CKD patients were not correlated with the concentrations of total-, high-density lipoprotein-, or low-density lipoprotein-cholesterol (Supplementary Fig. 14b). In contrast, serum retinol levels were positively correlated with RBP4 levels (Fig. 8f upper panel). Retinol and

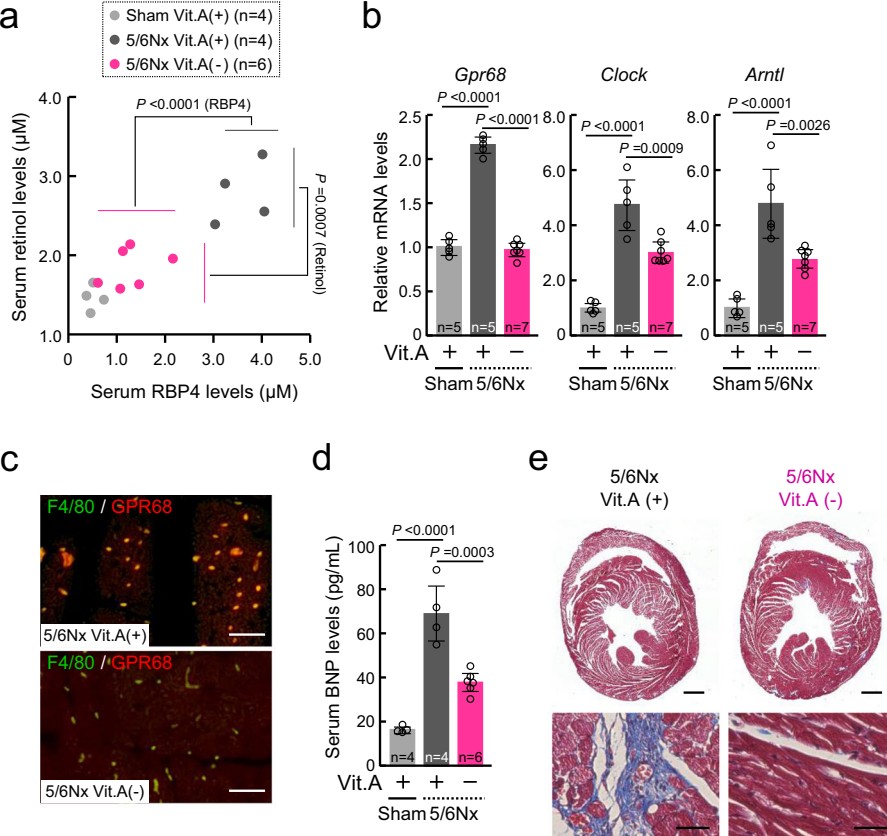

**Fig. 7 Vitamin A-free diet reduces GPR68 expression in monocytes in 5/6Nx mice. a** Correlation between serum retinol and RBP4 levels in 5/6Nx mice fed with normal or vitamin A-free diet. **b** Dietary deficiency of vitamin A suppresses the upregulation of *Gpr68*, *Clock*, and *Arntl* mRNA levels in circulating monocytes. The mean value of the Sham group was set as 1.0. **c** Double immunofluorescence labeling of GPR68 with F4/80 in the ventricle slices prepared from 5/6Nx mice fed with normal or Vitamin A-free diet. High-GPR68-expressing cells are double-labeled (yellow) with F4/80. The scale bar indicates 50 μm. **d** Serum BNP concentrations in Sham and 5/6Nx mice fed with normal or vitamin A-free diet. **e** Dietary deficiency of vitamin A ameliorates CKD-induced cardiac fibrosis. The Masson's trichrome staining show tissue fibrosis in blue. Scale bars indicate 1 mm (upper panel) and 50 μm (lower panel). For all panels, graphs show the mean ± SD of individual mice in independent experiments. Statistical significance was determined using one-way ANOVA with Tukey–Kramer post hoc tests (**a**, **b**, **d**). Numbers and *P*-values are shown in each graph. Source data are provided as a Source Data file.

RBP4 levels in the serum of CKD patients were both significantly higher than those in healthy subjects ($P < 0.05$) and they also increased depending on the serum creatinine levels (Fig. 8f lower panel). This suggested that serum retinol and RBP4 levels increase with deteriorating renal function. We therefore incubated human primary monocytes in the media supplemented with 20% serum of each individual CKD patient and measured the mRNA levels of *GPR68*. Induction of mRNA expression of *GPR68*, *TNFα*, and *IL-6* in human primary monocytes was detected when cells were incubated in 20% serum media (Fig. 8g). The mRNA expression levels of *GPR68*, *TNFα*, and *IL-6* in monocytes significantly increased depending on the serum retinol levels ($P < 0.01$, Fig. 8g). The production of TNFα and IL-6 from human primary monocytes was also increased by incubation with serum from CKD patients. Similarly, the expression of *CLOCK* and *ARNTL* in monocytes was increased by treatment with CKD patient serum (Fig. 8h). The mRNA expression levels of *CLOCK* and *ARNTL* also significantly increased, depending on the serum retinol levels ($P < 0.01$). The reported normal range of the serum BNP level is under 40 pg/mL, whereas heart failure can be diagnosed when the BNP level exceeds 100 pg/mL[49]. Monocytic expression of *GPR68* mRNA and serum retinol levels in CKD patients with a BNP level >100 pg/mL were significantly higher than those in healthy subjects with a BNP level under 40 pg/mL ($P < 0.05$ for GPR68 mRNA, $P < 0.01$ for retinol, Fig. 8i). The

levels of *GPR68* mRNA and serum retinol also increased in CKD patients with a BNP level between 40 and 100 pg/mL. Therefore, the accumulation of retinol and RBP4 in the serum during chronic renal failure induces GPR68 expression in monocytes. This may underlie the cardiac complications in CKD patients.

## Discussion

In this study, we found that high-GPR68-expressing monocytes infiltrated into the heart ventricle of 5/6Nx mice and they played an essential role in the exacerbation of cardiac inflammation and fibrosis (Fig. 9). Due to the interaction of organs, functional failure of one organ can lead to the dysfunction of others[50]. CKD is characterized as an independent risk factor for cardiovascular morbidity and mortality[51,52]. Extracellular volume expansion and activation of the RAAS are common phenomena in patients with CKD, which increase blood pressure, leading to increased cardiac load, ventricular hypertrophy, and heart failure[53,54]. On the other hand, the circadian machinery affects many circulatory functions by regulating RAAS, sympathetic nerve activity, blood pressure, pulse, and cardiac output; therefore, alteration of the transcriptional activity of CLOCK/ARNTL likely alters the functions of the circulatory system[13,15,22]. In addition to these cardiovascular problems, the infiltration of monocytes is often observed in the heart of CKD patients[55,56]. Our present study also revealed the

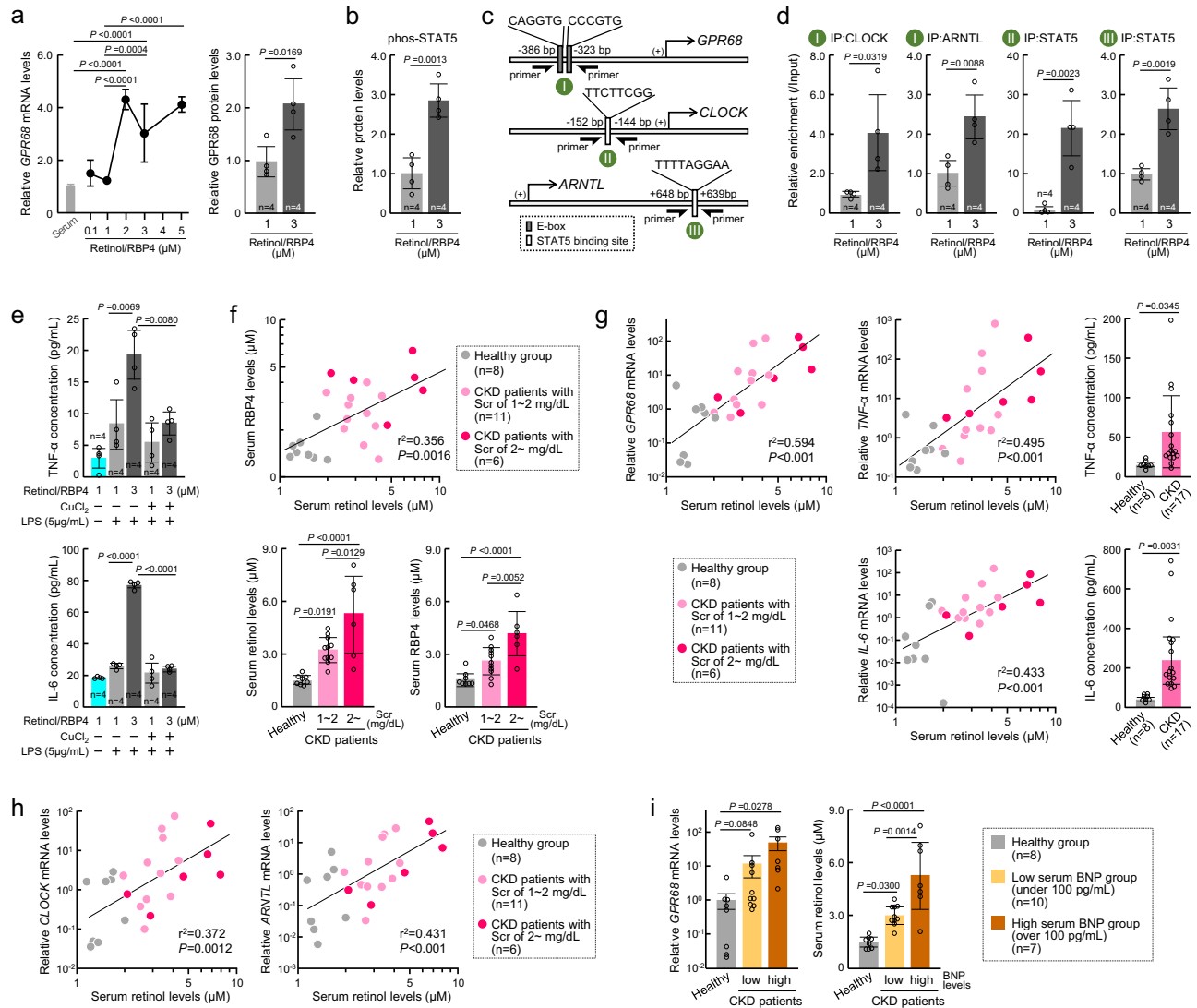

**Fig. 8 Serum from CKD patients induces GPR68 expression in human monocytes. a** Dose-dependent increase in the expression of *GPR68* mRNA (left) and its protein (right) in human primary cultured monocytes by treatment with retinol/RBP4 for 24 h. **b** The phosphorylation state of STAT5 in human primary cultured monocytes after treatment with retinol/RBP4. **c** Schematic representation of the upstream region of the human *GPR68*, *CLOCK*, and *ARNTL* genes. The circled letters (green circles) indicate the location on the gene where each of the different primer sets localize. **d** The binding of CLOCK, ARNTL, and STAT5 to regions, described in **c**, in human (healthy) primary cultured monocytes after treatment with retinol/RBP4 for 24 h. The primer sets used in qPCR are illustrated in **c**. **e** LPS-stimulated release of TNFα and IL-6 from human primary cultured monocytes after treatment with retinol/RBP4 for 24 h. **f** Serum retinol and RBP4 levels in CKD patients. The upper panel shows relationship between retinol and RBP4 levels. Lower panels show these levels stratified by serum creatinine levels (Scr). **g**, **h** Relationship retinol levels to mRNA expressions of *GPR68*, *IL-6*, *TNFα*, *CLOCK*, or *ARNTL* in human primary cultured monocytes incubated in media containing 20% serum collected from healthy subjects and CKD patients stratified by Scr. The release of TNFα and IL-6 from human primary cultured monocytes were assessed after incubation in the media containing 20% serum for 24 h. **i** Comparison of the values stratified by BNP concentrations. Left panel shows the expression levels of *GPR68* mRNA in human primary cultured monocytes incubated same as in **g**. Right panel shows serum retinol levels. For panel **f**–**i**, each plot shows a value obtained using an individual human serum. For all panels, graphs show the mean ± SD in independent experiments (**a**, **b**, **d**, **e** n = 5 for each group, **f**–**i** shown in each graph). Statistical significance was determined using one-way ANOVA with Dunnett's post hoc tests (**a**), two-tailed Student's *t*-tests (**a**, **b**, **d**, **g**), or one-way ANOVA with Tukey–Kramer post hoc tests (**e**, **i**). Numbers and *P*-values are shown in each graph.

monocytic expression of *Gpr68* in 5/6Nx wild-type mice activated by CLOCK/ARNTL, whose transcriptional activity was increased through the increase in serum retinol and RBP4 levels. Consequently, the monocytic expression of *Gpr68* was not induced in 5/6Nx *Clk/Clk* mice and they exhibited attenuation of cardiac inflammation and fibrosis, although they also had high blood pressure and RAAS activation. Cardiac infiltration of high-GPR68-expressing monocytes may play a major role in CKD-induced inflammation and fibrosis in the heart.

Several previous studies demonstrate that mice with a mutated *Clock* gene exhibit heart failure due to alteration of circulatory functions[11,15]. The circadian clock-related cardiovascular disorders are closely related to aging, cardiomyocyte hypertrophy, and accompanying decrease in ejection fraction. In contrast to these previous reports, we found that heart failure in 5/6Nx mice was associated with the continuous increase in the expression levels of *Clock* and *Arntl* in circulating monocytes. As 5/6Nx-induced renal inflammation was alleviated in *Clk/Clk* mice[24],

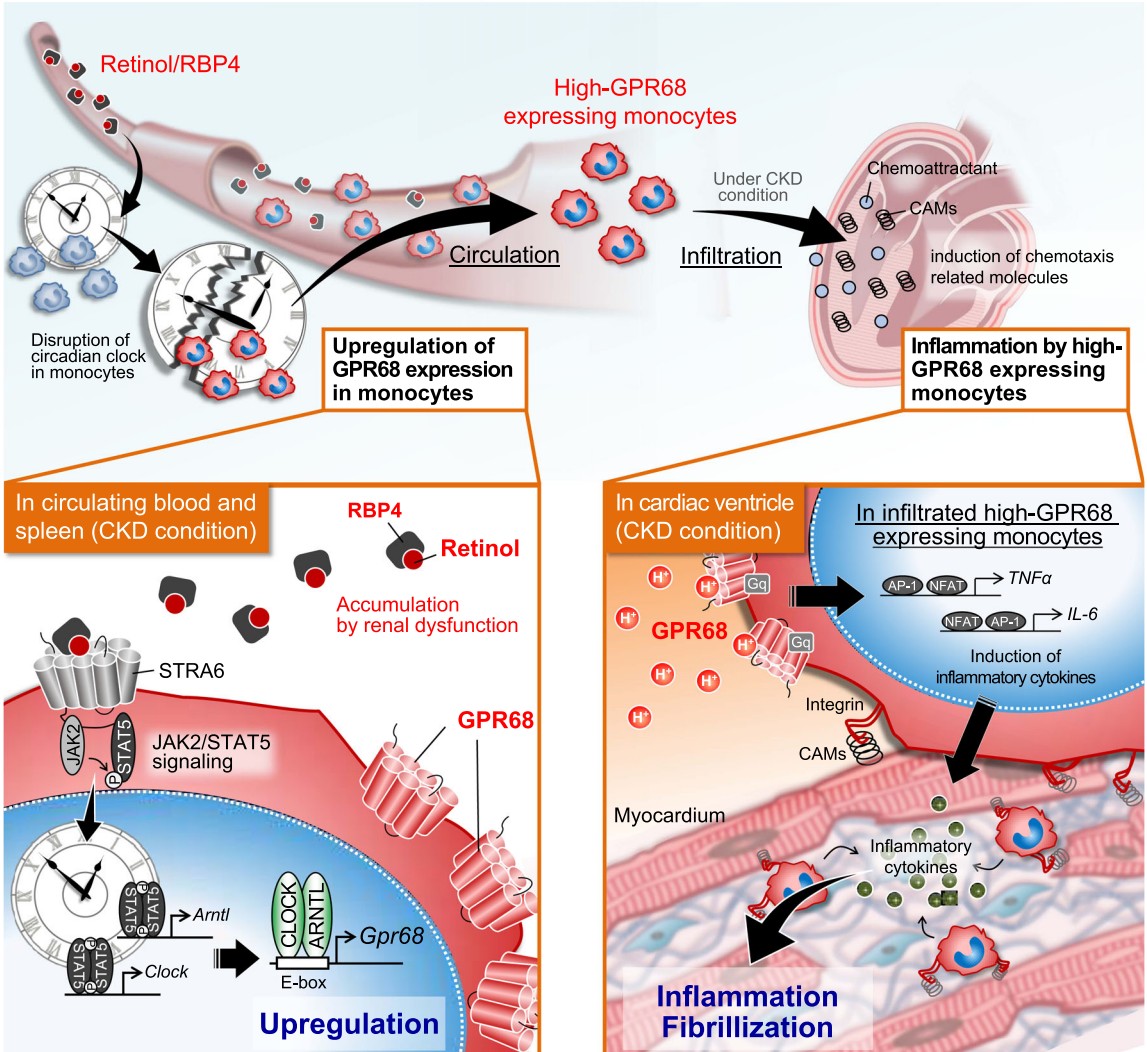

**Fig. 9 Schematic diagram indicating the retinol/RBP4-induced expression of GPR68 in circulating monocytes and their pathological role in the CKD-induced cardiac inflammation and fibrosis.** Serum levels of retinol and RBP4 were increased during chronic renal failure. The retinol-bound RBP4 activates JAK2/STAT5 signaling in circulating monocytes through STRA6 and induces the expression of GPR68 via activation of CLOCK/ARNTL. The high-GPR68-expressing monocytes infiltrate into the heart potential for producing inflammatory cytokines and their cardiac infiltration exacerbates inflammation and fibrosis.

CLOCK protein may promote the CKD-induced inflammatory response. The difference between previous studies and ours suggests that maintenance of normal *Clock* gene function is important to prevent the onset of heart failure. In addition, 5/6Nx mice exhibited increased serum BNP levels and cardiac fibrosis without reduced ejection fraction. These symptoms are similar to heart failure with preserved ejection fraction (HFpEF), which may represent up to half of the heart failure population[57,58]. When compared with heart failure with reduced ejection fraction, outcomes in HFpEF have not improved, highlighting the need for effective therapies for this condition. The hallmark of HFpEF is left atrial dilation, which is often accompanied by left ventricular hypertrophy and increased stiffness[59]. Recent studies reported the roles of cardiac inflammation and fibrosis in the exacerbation of HFpEF[60,61]. We observed monocytic expression of GPR68 and cardiac fibrosis in mice within 8 weeks after the 5/6Nx operation due to the decrease in their survival rate associated with uremia, decline in the glomerular filtration rate, and high urea nitrogen levels[62,63]. Longer observation of cardiac functions in 5/6Nx mice may make it possible to clarify the role of monocytic clock genes in the pathogenesis of CKD-induced inflammation and fibrosis in

the heart. This may also help in understanding the pathology of HFpEF and aid in the development of therapeutic strategies.

Monocytes are not homogenous and comprise subsets with distinct phenotypes[64,65]. In the present study, we found that ~90% of cardiac-infiltrated monocytes in 5/6Nx mice expressed GPR68. This G protein-coupled receptor functions as a proton sensor that is activated by an acidic extracellular pH through the protonation of histidine residues of the receptors and induces several downstream G protein pathways[37,66]. Under natural conditions, high-GPR68-expressing GPR68 monocytes released TNFα and IL-6 in response to LPS stimuli, but the LPS-stimulated release of inflammatory cytokines was further increased under acidic conditions. During acute myocardial ischemia, a decreasing pH is observed in the infarcted area, which is thought to facilitate the inflammatory responses[67,68]. However, the cardiac-infiltrated high-GPR68-expressing monocytes in 5/6Nx mice may have been activated by factors rather than the decrease in pH, because it is unlikely to cause myocardial ischemia and local acidosis during chronic renal failure. GPR68 is also expressed in mammalian small-diameter blood vessels, which sense shear stress[69]. Increased blood flow leads to acute

vasodilation, whereas long periods of increased blood flow cause structural remodeling, which causes outward growth and increases the diameter of vessel walls[69]. GPR68-knockout mice have disrupted vasodilatory responses to increased blood flow[69], suggesting that the G protein-coupled receptor mediates flow-mediated vasodilation and remodeling. The high-GPR68-expressing monocytes that infiltrated into the ventricle of 5/6Nx mice may have been exposed to shear stress around blood vessels and were activated to produce inflammatory cytokines.

The major known functions of the spleen are the removal of aging erythrocytes, recycling of iron, elicitation of immunity, and supply of erythrocytes after hemorrhagic shock[70]. Monocytes are released from the bone marrow into the peripheral blood, where they circulate for several days before seeding tissues[71,72]. The spleen also functions as a reservoir of monocytes and releases them into the peripheral blood in response to inflammation-associated diseases, including cardiovascular disorders. Therefore, removal of the spleen prevents the invasion of monocytes into perilesional tissues[73]. Indeed, splenectomy suppresses the increase in the number of circulating monocytes, resulting in suppression of their infiltration into the heart ventricle of 5/6Nx mice. In a mouse model of myocardial ischemic injury, depletion of circulating monocytes by splenectomy attenuated their recruitment to the infarcted myocardium, accompanied by suppression of inflammation and collagen deposition[71]. Furthermore, angiotensin II-induced hypertension and cardiac fibrosis were suggested to be associated with the recruitment of monocytes/macrophages from the spleen into the heart[74]. Consequently, the cardiac infiltration of monocytes may play a role in the pathophysiology of CKD-induced inflammation and fibrosis in the heart.

Patients with CKD have reduced urine production and lose blood-filtering function, leading to increasing concentrations of metabolic waste and uremic toxins in the blood and tissues[75]. During the progression of CKD, the serum retinol levels are also increased due to alteration of the hepatic metabolism pathway[26,27]. Monocytic expression of *Gpr68* in 5/6Nx wild-type mice was activated by CLOCK/ARNTL whose expression and transcriptional activity were enhanced through the increase in serum retinol levels, and the retinol-induced expression of CLOCK and ARNTL was mediated via STRA6. Recent studies demonstrated that STRA6 not only functions as a retinol-bound RBP4 transporter but also as a ligand-activated cell surface signaling receptor, which, upon binding of the retinol-bound RBP4, activates JAK2/STAT5 signaling, inducing the expression of STAT target genes[43]. Higher expression levels of STRA6 are observed in the brain, kidneys, spleen, testes, and female genital tract[76]. Indeed, increased expression of CLOCK protein is also detected in the kidneys but not in the liver of 5/6Nx mice[24,26]. Therefore, retinol-induced expression of CLOCK protein may be dependent on the levels of STRA6. The altered expression of clock genes is also observed in other CKD model animals[25]. The expression rhythms of clock genes were attenuated in the kidneys of mice fed an adenine-containing diet. Disruption of the circadian clock was suggested to further accelerate the progression of CKD. As CKD patients often develop sleep disorders, altered clock gene expression may also be induced in humans.

After intracellular permeation, retinol binds to RBP1 and is converted to retinal by alcohol dehydrogenase, and then to retinoic acid by retinal dehydrogenases encoded by *aldehyde dehydrogenase 1a1* (*Aldh1a1*), *Aldh1a2*, and *Aldh1a3* (Supplementary Fig. 15a). These then activate RAR and RXR to induce the expression of their target genes. However, this RAR/RXR-mediated pathway was unlikely involved in the monocytic expression of *Gpr68* in 5/6Nx mice (Supplementary Fig. 10). The expression levels of *Aldh1a* mRNA in the kidney of 5/6Nx mice were

significantly higher than those in the heart and monocytes (Supplementary Fig. 15b). High expression of *Aldh1a* in the kidney of 5/6Nx mice may cause the rapid conversion of retinol to retinoic acid, and then further induce the activation of the RAR/RXR-mediated pathway rather than JAK2/STAT5 signaling.

Retinol and RBP4 levels in CKD patients were both significantly higher than those in healthy subjects. Retinol is derived from the dietary intake of both retinol and pro-vitamin A[45,46], whereas RBP4 is mainly synthesized in the liver and secreted into the circulation[45,46,77]. The kidneys are the main site of RBP4 catabolism, which functions in the increase in RBP4 levels during chronic renal failure[26,78]. Hepatic secretion of RBP4 is also increased by retinol[79,80]. Indeed, the administration of retinol to normal mice increased serum RBP4 levels (Supplementary Fig. 12a). There was a correlation between retinol and RBP4 levels in the human serum samples. As expression levels of GPR68 and inflammatory cytokines in the human monocytes also increased depending on serum retinol levels, increases in serum levels of retinol and RBP4 may be a factor for CKD-induced inflammation and fibrosis in the heart.

Feeding of 5/6Nx mice with a vitamin A-free diet, which was initiated at 8 weeks after nephrectomy, prevented the increase in serum BNP levels and cardiac fibrosis without suppression of renal dysfunction. On the other hand, our previous study demonstrated that initiation of dietary vitamin A deficiency in 5/6Nx mice at 4 weeks after nephrectomy suppressed renal dysfunction with elevation of serum urea nitrogen levels[26]. This discrepancy may be due to the difference in the degree of renal impairment at the initiation of vitamin A deficiency. Serum levels of urea nitrogen and creatinine in 5/6Nx mice begun to increase around 6 weeks after nephrectomy (Supplementary Fig. 1a). Renal function of 5/6Nx mice might have been irreversibly impaired at 8 weeks after nephrectomy. Therefore, the initiation of vitamin A deficiency in 5/6Nx mice after the exacerbation of renal dysfunction would be late to suppress the increase in serum urea nitrogen levels. In contrast to kidney, heart failure of 5/6Nx mice might be still reversible at 8 weeks after surgical nephrectomy. The inflammation response in the kidney of 5/6Nx mice is probably caused by the activation of RAR/RXR-mediated pathway, but their cardiac inflammation is dependent on the infiltration of GPR68-expressing monocytes. The mechanistic difference in the inflammation response between kidney and heart may contribute to the progression rate of their impairment.

Although tissue infiltration of monocytes was observed in the heart and kidney of 5/6Nx mice, no notable infiltration of monocytes was detected in the liver (Fig. 2a and Supplementary Fig. 16d). This organ-dependent difference in monocyte infiltration may be associated with distinct expression levels of adhesion molecules (Supplementary Figs. 5 and 16a–c). The induction of VCAM1 expression in the heart and kidney of 5/6Nx mice was not affected by the administration of retinol (Supplementary Fig. 12d) or dietary deficiency of vitamin A (Supplementary Fig. 17b), suggesting that the induction of adhesion molecules was not due to changes in the retinol levels, but instead due to high blood pressure and RAAS activation[81,82]. In contrast to the heart, GPR68-expressing cells were undetectable in the kidney of 5/6Nx mice (Supplementary Fig. 18) even though the renal infiltration of monocytes was observed (Supplementary Fig. 19a). There are several possible reasons for not detecting GPR68-expressing monocytes in the kidney of 5/6Nx mice. Kidney-infiltrating GPR68-expressing monocytes in 5/6Nx mice may be eliminated by excessive activation of apoptotic signaling. High serum retinol levels in 5/6Nx mice activate renal caspase activity, leading to apoptotic cell death[26]. On the other hand, transfroming groth factor-β1 (TGFβ1) produced from the injured kidney may induce the differentiation of infiltrated monocytes. The expression and

production of TGFβ1 were increased in the kidneys but not in the heart of 5/6Nx mice (Supplementary Fig. 19b). The expression of *Gpr68* mRNA in primary cultured monocytes was increased by incubation in the media containing 10% serum collected from 5/6Nx mice, but 5/6Nx serum-induced expression of *Gpr68* mRNA was inhibited by TGFβ (Supplementary Fig. 19c). Treatment with TGFβ also induced the differentiation of Ly6C+ monocytes into Ly6C− cells (Supplementary Fig. 19d). Therefore, kidney-infiltrated GPR68-expressing monocytes may differentiate into different subtypes of macrophages. Taken together, cardiac infiltration of GPR68-expressing monocytes in 5/6Nx mice may be associated with organ-dependent differences in the expression of adhesion molecules and tissue-specific differentiation systems.

In conclusion, we identified a mechanism underlying the cardiac complications in mice with chronic renal failure. Serum accumulation of retinol induces the CLOCK/ARNTL-mediated transactivation of GRP68 in circulating monocytes and their infiltration into the heart ventricle exacerbates inflammation and fibrosis. Increased serum retinol levels in CKD patients are also sufficient for inducing the expression of GPR68 and inflammatory cytokines in human monocytes. As lipophilic non-dialyzable compounds, including retinol, often accumulate in CKD patients due to a lack of elimination through renal metabolism and dialysis[83], development of a strategy to suppress retinol accumulation will be useful to prevent the cardiac complications of CKD.

## Methods

**Animals and treatments.** All animal experiments were conducted in accordance with the Guidelines for Animal Experiments of Kyushu University and were approved by the Institutional Animal Care and Use Committee of Kyushu University (approved protocol ID #A30-061). *Clock* mutant mice (C57BL/6J-ClockmlJt/J) were purchased from the Jackson Laboratory (Bar Harbor, ME, USA) and backcrossed to wild-type Jcl: ICR mice (Charles River Laboratory Japan, Inc.; Yokohama, Japan) for more than eight generations to improve breeding and off-spring care. Male ICR mice were housed in a light-controlled room (lights on from ZT0 to ZT12) at 24 ± 1 °C and 60% ± 10% humidity, and the animals had free access to water and normal pelleted diet (positive control diet groups contained 4000 IU of vitamin A/kg) or a vitamin A-free AIN-93G pelleted diet (Oriental Yeast Co., Ltd). Mice were synchronized to the light/dark cycle for 2 weeks before surgery. For splenectomy, the abdominal wall of the mouse (5 weeks old) was opened by making a left subcostal minimal incision under isoflurane anesthesia (Baxter, Deerfield, IL, USA). After the splenic arteries and veins were ligated at the splenic hilum, the spleen was removed. Sham mice were subjected to laparotomy on the same days as the splenectomy. Two weeks after splenectomy, splenectomized and Sham mice were subjected to 5/6Nx as follows. CKD model mice were generated by two-step 5/6Nx at 6 or 7 weeks of age. At the first surgery, two-thirds of the left kidney was removed by cutting off both poles. Seven days later, the right kidney was removed in entirety. After the operations, the mice were housed for 8 weeks until CKD development. Sham mice were subjected to laparotomy on the same days as the 5/6Nx surgeries. To downregulate GPR68, 5/6Nx mice were injected with control or anti-*Gpr68* siRNA (40 μg/week; Life Technologies, Tokyo, Japan) encapsulated in HVJ-E (ISHIHARA SANGYO, LTD, Japan) via the tail vain. The siRNAs were injected every week from 4 to 8 weeks after the 5/6Nx operation. The siRNA oligonucleotide sequences were as follows: *Gpr68* siRNA sense 5′-GCAUCCUCCUC UAUGAGAACAUUUA-3′ and antisense 5′-UAAAU GUUCUCAUAGAGGAGGAUGC-3′; control siRNA sense, 5′-GCACUCCUCUA UGAGCAAUAUCUUA-3′ and antisense 5′-UAAGAUAU UGCUCAUAGAGGA GU-GC-3′. To upregulate GPR68 expression in normal mice, naive ICR mice were injected with control (pcDNA3.1) or GPR68-expressing plasmid (40 μg; OriGene Technologies, Inc., Rockville, MD, USA) encapsulated in HVJ-E (ISHIHARA SANGYO, LTD) via the tail vain. Each plasmid was administered every 3 days for 2 weeks. Naive ICR mice were also intraperitoneally administered 100 μL of retinol (25 mM) or vehicle (dimethyl sulfoxide) every 3 days for 2 weeks.

**Cell culture and treatment.** Mouse primary monocytes were prepared from the blood of 6-week-old ICR littermates using the MACS Monocyte Isolation Kit (Miltenyi Biotec Ltd, Bisley, GB). Cells were cultured in RPMI1640 medium supplemented with 0.5% penicillin/streptomycin and 10% mouse serum (prepared from Sham or 5/6Nx wild-type mice) under a 5% CO₂ environment at 37 °C. To downregulate STRA6, mouse primary cultured monocytes were exposed to control or anti-*Stra6* siRNA (Life Technologies) encapsulated in HVJ-E (ISHIHARA SANGYO, LTD) for 24 h. The siRNA oligonucleotide sequences were as follows: *Stra6* siRNA sense 5′-GCUGCUGUCUUUGUGGUCCU-CUUCA-3′ and

antisense 5′-UGAAGAGGACCACAAAGACAGCAGC-3′; control siRNA sense, 5′-GCUGUCUUUGUGUUGCCCUUCGUCA-3′ and antisense 5′-UGACGAAG GGCAACACAAAGACA-GC-3′. Lenti-X 293T (CLN632180, Takara Bio Co., Ltd, Osaka, Japan) were maintained in Dulbecco's modified Eagle's medium supplemented with 10% fetal bovine serum (FBS) and 0.5% penicillin–streptomycin. Human primary monocytes (Human CD14+ monocytes from peripheral blood from a single donor, Takara Bio Co., Ltd) were cultured under a 5% CO₂ environment at 37 °C in BIOTARGET media (Biological Industries Ltd, Beit-Haemek, Israel) supplemented with 20% serum (collected from healthy subjects or CKD patients) and 0.5% penicillin/streptomycin. The handling of human serum samples and experimental protocol were approved by the research ethics committees of Nagasaki Kamigoto Hospital (approved protocol ID #26-11) and Kyushu University Hospital (approved protocol ID # 757-00), and are complied with all relevant ethical regulations. We received the written informed consent from all study subjects.

**Construction and transfection of miRNA.** The *Gpr68*-targeting sequence (5′-CCTCCTCTATGAG-AACATTTA-3′) was designed by BLOCK-iT™ RNAi Designer (Thermo Fisher Scientific, Tokyo, Japan). The target sequence was inserted into pcDNA6.2™-GW/miR (Thermo Fisher Scientific) using BLOCK-iT™ Pol II miR RNAi Expression Vector Kits (Thermo Fisher Scientific). The sequence of anti-*Gpr68* miRNA containing the flanking region of the plasmid was amplified by PCR and then inserted into the Mlu1/Cla1 site downstream of the H1 promoter in lentiviral plasmids pLVTHM (Addgene, Watertown, MA, USA)[84]. The control miRNA sequence was also amplified from pcDNA6.2™-GW/miR and inserted into pLVTHM by the same method. The miRNA-expressing lentiviral plasmids, Lentiviral High Titer Packaging Mix (Takara Bio Co., Ltd), and TransIT-293 Transfection Reagent (Takara Bio Co., Ltd) were added to the media of Lenti-X 293T cells, and incubated for 48 h. After incubation, the virus-containing medium was collected and concentrated using a Lenti-X Concentrator (Takara Bio Co., Ltd). Monocytes (1.0 × 10⁶) were prepared from the spleen of ICR male mice. Cells were incubated in RPMI1640 media containing lentivirus in the presence of 5 μg/mL of polybrene at 37 °C. Following collection and resuspension in saline, cells were injected into Splx-5/6Nx mice via the tail vein.

**Quantitative RT-PCR.** Total RNA was extracted using RNAiso (Takara Bio Co., Ltd, Osaka, Japan) or a QIAGEN RNeasy Mini kit (QIAGEN Sciences, Valencia, CA). cDNA was synthesized using a ReverTra Ace qPCR RT kit (Toyobo, Osaka, Japan) and amplified by PCR. Real-time PCR analysis was performed on diluted cDNA samples using the THUNDERBIRD SYBR qPCR Mix (Toyobo) with the 7500 Real-time PCR system (Applied Biosystems, Foster City, CA, USA). Data were normalized using 18s and β-actin mRNA as controls. Primer sequences are listed in Supplementary Table 6.

**Western blotting.** Cells were homogenized in CelLytic MT Cell Lysis Reagent (Sigma-Aldrich). Nuclear proteins were prepared using LysoPure™ Reagent (Sigma-Aldrich). Samples were separated by SDS-polyacrylamide gel electrophoresis and transferred to polyvinylidene difluoride membranes. The membranes were reacted with primary antibodies against mouse GPR68 (1 : 1000; CSB-PA060199, CUSA-BIO, TX, USA), ARNTL (1 : 1000; ab-93806, Abcam, Cambridge, UK), CLOCK (1 : 1000; ab-3517, Abcam), phosphorylated STAT5A/B (Y694/699) (1 : 1,000; 9359; Cell Signaling, MA, USA), STAT5A/B (1 : 1000; sc-835; Santa Cruz Biotechnology, TX, USA), STRA6 (1 : 1000; NBP1-04242, Novus Biologicals, CO, USA), TNF-α (1 : 1000; BL-506307, BioLegend, CA, USA), IL-6 (1 : 1000; BL-504507, BioLegend), RBP4 (1 : 1000; ab-109193, Abcam), and ACTB (SC-47778, Santa Cruz Biotechnology). The immunocomplexes were reacted with anti-guinea pig, anti-mouse or anti-rabbit IgG secondary antibody, and Chemi-Lumi One reagent (Nacalai Tesque, Inc., Kyoto, Japan). The membranes were photographed and the density of each band was analyzed using an ImageQuant LAS 3000 mini (Fuji Film, Co., Ltd, Tokyo, Japan). All antibodies were diluted in Can Get Signal Immunoreaction Enhancer Solution (Toyobo). The calibration curve for calculating RBP4 concentrations was made by the quantified values of recombinant RBP4 protein levels (mouse; R&D systems, Inc., Minneapolis, MN, USA. Human; BioLegend), which was measured by western blotting as described above.

**Histochemical staining.** Eight weeks after the 5/6Nx operation, hearts were removed from Sham and 5/6Nx mice, and then placed in 4% paraformaldehyde in phosphate-buffered saline (PBS) for 12 h. The fixed hearts were paraffin-embedded and stained with Masson's trichrome, followed by rinsing in 15% or 30% sucrose and snap-freezing. The blue-stained fibrotic regions were measured quantitatively using a computer-assisted image analyzer under a Keyence BZ-9000 fluorescence microscope at ×10 or ×20 magnification.

**Immunofluorescence histochemical staining.** Immunofluorescence histochemical staining was performed using frozen heart, kidney, and liver sections. Cryostat cardiac sections fixed with 4% paraformaldehyde were blocked in solution containing 10% FBS and 0.1% Triton X-100 for 1 h at 4 °C, followed by incubation with antibodies against GPR68 (Novus Biologicals), VCAM1 (Abcam), or F4/80 (BioRad, Hercules, CA, USA) at 4 °C for 24 h. After washing, the sections were

incubated with a fluorescent secondary antibody (Alexa 488 or Alexa 546, Jackson ImmunoResearch, Laboratories, West Grove, PA) at 4 °C for 2 h. The slices were mounted using Vectashield hard-set mounting medium with 4′,6-diamidino-2-phenylindole (Vector Laboratories, Burlingame, CA, USA). Visualized images were scanned using a BZ-9000 instrument (Keyence, Osaka, Japan). All antibodies were diluted in Can Get Signal Immunostain Immunoreaction Enhancer Solution (Toyobo).

**ChIP analysis**. Mouse or human monocytes were incubated with 4% formaldehyde at 4 °C for 20 min. The cross-linked cells were homogenized in cell lysis buffer (10 mM HEPES-KOH pH 7.9, 1.5 mM MgCl₂, 10 mM KCl, 0.5 mM dithiothreitol, and 0.5% NP-40) and incubated for 10 min on ice. Nuclei were obtained by centrifugation at $1400 \times g$ for 5 min. Nuclear lysis buffer was added to the pellets (50 mM Tris-HCl pH 8.1, 10 mM EDTA, and 1% SDS) and sonicated on ice. The lysates were centrifuged at $10,000 \times g$ for 10 min. The supernatants were incubated with antibodies against ARNTL (ab-93806, Abcam), CLOCK (ab-3517, Abcam), STAT5A/B (sc-835; Santa Cruz Biotechnology), or rabbit IgG (MBL) in dilution buffer (purified water with 16.7 mM Tris-HCl, 167 mM NaCl, 1.2 mM EDTA, 1.1% Triton X-100, 10% SDS). DNA was purified using a QIAquick PCR purification Kit (Qiagen) and was amplified by PCR for E-box and STAT5-binding elements in the 5′-flanking region of the mouse *Gpr68*, *Clock*, and *Arntl* genes. Primer sequences for amplification are shown in Supplementary Table 7. THUNDERBIRD SYBR qPCR Mix (Toyobo) with the 7500 Real-time PCR system (Applied Biosystems) was used to quantify the products. All data were normalized to the PCR products of input DNA. The quantitative reliability of PCR is evaluated by kinetic analysis of the amplified products, to ensure that signals were only from the exponential phase of amplification. This analysis also proceeded in the absence of an antibody or in the presence of rabbit IgG as negative controls. Ethidium bromide staining detected no PCR products in these samples.

**Echocardiography**. We performed transthoracic ultrasound cardiography on mice at 8 weeks after the 5/6Nx operation under anesthesia according to the established method[85]. Two-dimensional targeted M-mode tracings were recorded using a Nemio GX image analyzing system (SSA-580A, Toshiba Medical Systems, Japan). The %EF was calculated using Pombo's method. To estimate cardiac systolic function, the percent fractional shortening (%FS) was calculated as follows: %FS = [(LVIDd − LVIDs)/LVIDd] × 100.

**Isolation of ventricular-derived cells**. Eight weeks after the 5/6Nx operation, the heart was exposed and perfused with 10 mL of PBS from the left ventricle. After removing the atria, the ventricles were digested in PBS containing 500 µg/mL of collagenase type II (FUJIFILM Wako Pure Chemical Corporation, Osaka, Japan), 200 µg/mL of CaCl₂ (Nacalai Tesque, Kyoto, Japan), 0.05% trypsin (Sigma-Aldrich), and 10% FBS at 37 °C for 10 min with agitation. Digested samples were further dissociated by passing them through a 23-gauge needle three times. After treating with Red Blood Lysis buffer (BioLegend, CA, USA), isolated cells were filtered through a 40 µm strainer.

**Flow cytometry**. After treatment of isolated cardiac cells with mouse TruStain FcX (BL-101320, BioLegend), samples were stained with anti-F4/80-APCfire™ (BL-123108, BioLegend), anti-CD11b-PE (BL-101208, BioLegend), anti-Ly6C-APC (BL-128016, BioLegend), anti-Ly6C-FITC (BL-128006, BioLegend), and/or anti-GPR68 (CSB-PA060199; CUSABIO). Cells stained with anti-GPR68 were further stained with anti-rabbit IgG-AF647 (ab-150107, Abcam). To detect IL-6- and TNFα-expressing cells, cells were also incubated with 10 µg/mL of Brefeldin A (Funakoshi), followed by fixation with 2% paraformaldehyde and permeabilization with Intracellular Staining Permeabilization (BioLegend). After fixation and permeabilization, cells were stained for anti-IL-6-APC (BL-504507, BioLegend) and anti-TNF-α-APC (BL-506307, BioLegend). Dead cells were labeled by eFluor 780 viability dye (BD-565388, BD Biosciences, Erembodegem, Belgium). Stained cells were applied to flow cytometry (Aria III; BD Biosciences) and the results were analyzed using FlowJo (Tree Star).

**Microarray gene expression analysis**. Ventricle sections, kidney, liver, and splenic monocytes were collected 8 weeks after the 5/6Nx operation as described above. Total RNA was extracted using RNAiso (Takara). The quality of extracted RNA was analyzed using an Agilent 2100 Bioanalyzer (Agilent Technologies, Palo Alto, CA). The cRNA was amplified and labeled using a Low Input Quick Amp Labeling Kit (Agilent Technologies, Inc., Loveland, CO). Labeled cRNA was hybridized to a 44 K Agilent 60-mer oligomicroarray (Whole Mouse Genome Microarray Kit Ver.2.0). To identify upregulated genes, we calculated ratios (non-log scaled fold change) from the normalized signal intensities of each probe. We set criteria for up- or downregulated genes as follows: ratio > 1.5 and Z-score > 2 or ratio < 0.67 and Z-score < −2, respectively. Functional analysis of increased gene expression was performed using the KEGG database on the DAVID system and Gene Ontology analysis[41]. Raw microarray data were submitted to the Gene Expression Omnibus in NCBI (Cardiac ventricle: GSE150094, Kidney: GSE57799,

Liver: GSE35135, Monocytes: GSE140706). PBS with 10% FBS and 10 mM EDTA was used for washing of all cells and dilution of all antibodies.

**Quantification of TNFα, IL-6, angiotensin II, aldosterone, BNP, and blood urea nitrogen**. Plasma concentrations of TNFα, IL-6, angiotensin II, aldosterone, and BNP were measured using LEGENDMAX® ELISA Mouse/human TNFα (BioLegend), LEGENDMAX® ELISA Mouse/human IL-6 (BioLegend), Angiotensin II ELISA kit (Enzo Life Science, NY, USA), Aldosterone ELISA kit (Enzo Life Science), and Brain Natriuretic Peptide EIA Kit (RayBio, Shanghai, China), respectively. Serum blood urea nitrogen levels were measured using a manufactured kit (Wako Pure Chemical Industries Co. Ltd, Osaka, Japan).

**Measurement of retinol and retinoic acid concentrations**. Serum levels of retinol retinoic acid were measured by liquid chromatography/mass spectrometry analysis according to the previous report[86,87]. Up to 500 µL of serum was added to a disposable glass culture tube (16 mm × 150 mm), followed by 15 µL of the internal standard, 50 nM 4,4-dimethyl retinoic acid in acetonitrile (Sigma-Aldrich). A volume of 1 ml of 0.025 M KOH in ethanol was added and then vortex-mixed for 30 s. After the addition of 10 mL hexane, the samples were mixed further and then centrifuged for 3 min at 60 rev./min. The hexane layer containing neutral lipids, including retinol and retinyl esters, was removed from the samples. For the recovery of retinoic acid, 60 µL of 4 M HCl was added to the aqueous phase and the sample was vortex-mixed for another 30 s. After the addition of 10 mL of hexane, the hexane layer was removed as described above. The solvent was removed under a gentle stream of nitrogen with heating at 25–30 °C. The residue was dissolved in 100 µL of acetonitrile and placed in a deactivated glass autoinjector insert for analysis.

Retinol and retinoic acid were resolved using an AQUITY UPLC HSS PFP column (50 mm × 2.1 mm, 50 mm, p/n186005965, Waters, Milford, MA, USA) on an AQUITY UPLC H-Class XHCLQT0100 system (Waters), consisting of a vacuum degasser, binary pump, thermostatically controlled column compartment, thermostatically controlled autosampler, and a diode array detector. Mobile-phase A consisted of 0.1% formic acid/water; mobile-phase B consisted of 0.1% formic acid/acetonitrile. A linear gradient was generated at 1 ml/min: 0–0.3 min, 40% A, and 60% B; 0.3–3.5 min, 5% A, and 95% B; 3.5–4.0 min, 40% A, and 60% A (Waters application note). The injection volume was 10 µL. The column temperature was controlled at 25 °C and the autosampler compartment was set to 10 °C. An AQUITY TQ mass spectrometer controlled by Masslynx™ software (Waters) was operated in selected reaction monitoring mode. The retention times for retinol, retinoic acid, and the internal standard were 1.73, 1.68, and 2.23 min, respectively. Optimum positive Atmospheric Pressure Chemical Ionization (APCI) conditions included the following: nebulizer gas, 4; curtain gas, 6; collision gas, 6; nebulizer current, 3; and temperature, 400 °C. The multiple reaction monitor was set at a mass-to-charge ratio ($m/z$) of 286.4–92.9 $m/z$ for retinol, 300.4–283.3 $m/z$ for retinoic acid, and 329.1–151.1 $m/z$ for the internal standard. The spectra of retinol (Car# 95144, Sigma-Aldrich) and retinoic acid (Car# R2625, Sigma-Aldrich), which were used for the treatment of primary cultured monocytes and the standard curve to calculate sample concentrations, are shown in Supplementary Fig. 20.

**Statistics and reproducibility**. All statistical analyses were carried out using JMP® Pro 13.0.0 software (SAS Institute Japan, Tokyo, Japan). The significance of differences was analyzed by a one- or two-way analysis of variance and post hoc Dunnett's test or Tukey–Kramer's test among multiple groups, and the Student's $t$-test between pairs of groups. All statistical test used two-sided. A 5% level of probability ($P < 0.05$) was considered significant. The representative data shown as western blotting photograph and microscopy images are obtained from at least three independent experiments or three mice per group (for Figs. 2a, f, 3e, and 7e, and Supplementary Figs. 6d, 7c, d, f, 11d, 12d, e, g, 16c, d, 17b, and 18).

**Reporting summary**. Further information on research design is available in the Nature Research Reporting Summary linked to this article.

## Data availability

All data supporting the results of the present study are included in the article, either in the main figures or the Supplementary Information files. Microarray data were submitted to the Gene Expression Omnibus at the National Center for Biotechnology Information (Cardiac ventricle: GSE150094, Kidney: GSE57799, Liver: GSE35135, and Monocytes: GSE140706). Functional analyses of the microarray datasets were performed using KEGG database (https://www.genome.jp/kegg/kegg_ja.html) and Gene Ontology Recource (http://geneontology.org/). Any data supporting the findings of this study will be available from the corresponding author upon reasonable request. Source data are provided with this paper.

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

## Acknowledgements
This work was supported in part by a Grant-in-Aid for Scientific Research A (16H02636 to S.O.), Grant-in-Aid for Challenging Exploratory Research (17H06262 to S.O.), Grant-in-Aid for JSPS Fellows 18J20628 (to Y.Y.) from the Japan Society for the Promotion of Science, and Scientific Research B (18H03192 to N.M.) from Japan for the Promotion of Science. This research is supported by Platform Project for Supporting Drug Discovery and Life Science Research (Basis for Supporting Innovative Drug Discovery and Life Science Research (BINDS)) from the Japan Agency for Medical Research and Development (AMED) under Grant number JP20 and JP21 am0101091.

## Author contributions
Y.Y., N.M., and S.O. designed the research. Y.Y., N.M., T.N., K.H., H.K., T.I., H.T., A.T., M.K., M.N., and H.K. acquired the data. Y.Y., N.M., S.K., and S.O. drafted the manuscript.

## Competing interests
The authors declare no competing interests.
