## [Peer Review File · Nature Communications]

REVIEWER COMMENTS

Reviewer #1 (Remarks to the Author):

This well-written manuscript reports the negative effect of the circadian machinery during chronic kidney disease (CKD) on cardiovascular disease. The main mechanism was shown to be induction of Gpr68. These studies appear sound and add insight into mechanisms of CKD and its effects on heart disease.

The major concerns are those sections of the manuscript which correlate retinol status to Gpr58 and other sequelae of CKD and its effects on heart disease. The connection with retinol is weak for both CKD and heart disease. It is based on literature that may be unsound, correlative, and on tenuous extrapolations by the authors. The emphasis on retinol in the Abstract and in the later figures detract from an otherwise strong presentation.

Part of the rationale for this investigation was a report (Ref. 26) correlating increased RBP4 levels (not retinol levels) with CKD inducing heart disease. Although RBP4 is a carrier for retinol, functions of RBP4 also differ from those of retinol. Moreover, the reference cited did not use an assay for RBP4 that has been established as valid (PMID17294166). In addition, the reference concluded that CKD also correlates with age, BMI, and current smoking status. The method used to measure RBP4 in reference 26 reported a median value of 34 mg/dL. This is 10-fold higher than the concentration reported in normal humans. Another article (PMID: 17618858) reported that RBP4 correlates with obesity, type II diabetes, and glucose intolerance. Therefore, the claim that retinol correlates with CKD overlooks the many other correlates. The reference (26) did not establish a sound correlation between retinol and CKD. Nor does the present manuscript do so, or establish a causative relationship.

A case in point is this statement "... the abnormal increase in serum retinol levels in 5/6Nx wild-type mice increased retinol inflammation, further exacerbating the pathologies of CKD. In contrast, 5/6Nx-induced renal inflammation was attenuated in Clk/Clk mice²⁴, although they had high serum retinol levels". First, what is "retinol inflammation"? Second, doesn't this observation disconnect retinol from "retinol inflammation" from CKD caused heart disease?

The statement that "Serum retinol concentrations were measured as described previously⁷²" does not seem accurate. The innovation in the reference (ref 72) was an extraction protocol that improved recovery. The current manuscript used a different extraction protocol and did not provide other essential details verifying the efficacy of the retinol analysis. This is important because the authors' base major conclusions of the current report on an increase in retinol.

The retinol concentrations in the serum of the Jcl/ICR mice at ≥ 6 μM were much higher than in humans and other mouse strains, which are ≤ 1 to 2 μM , and lower than 1 μM in most tissues (except livers) of most mice and humans. This may be a technical problem, or reveal that this strain has unusual retinoid metabolism.

Fig. 6c shows an increase in mRNA of Gpr68 in RAW264.7 cells after treatment with high concentrations of retinol. Retinol does not bind with high affinity (even at 10 μM) to nuclear receptors. There must be intermediary events between retinol and an increase in Gpr68 mRNA and STAT5 protein binding to the upstream regions of the Clock and Arntl genes. Also, no mention was made to purifying retinol. Commercial retinol often is contaminated with up to 4% retinal and some retinoic acid. Because of the high retinol concentrations used, one of these potent retinoids or other contaminant might account for the activity observed.

Fig. 6g shows an increase with high retinol treatment of STAT binding to upstream regions of Clock and Arntl genes, but in RAW264.7 cells. The essential experiment should have been done with monocytes/macrophages harvested from WT mice and humans to show a direct effect of (purified) retinol on Gpr68 expression and STAT5 binding to the upstream regions of Clock and Arntl genes. Some experiments were done with monocytes from mice, which demonstrates the feasibility of performing the crucial experiments, while using lower concentrations of purified retinol.

Fig. 7e, which correlates serum retinol with creatinine levels in CKD patients, has only two data points for each of the high serum levels.

Fig. 7f entitled "Correlation of serum retinol levels of CKD patients with mRNA expression of GPR68, IL-6, or TNF α in THP-1 cells", was not only a correlation, but the correlation was made with an established cell line, not with cells from CKD patients, nor in primary cells from mice.

Contrary to the statement, Stra6 does not act as a receptor for retinol. It binds only RBP4. This is more than a semantic difference.

The statement "Our present study also revealed the monocytic expression of Gpr68 in 5/6Nx wild-type mice activated by CLOCK/ARNTL, whose transcriptional activity was increased through the increase in serum retinol levels" is not substantiated by the data because: 1) no evidence of this was shown in primary monocytes from humans or mice; the phenomenon was induced by high retinol concentrations that were not shown to occur in CKD patients.

The model in 7G is not justified by the data.

Reviewer #2 (Remarks to the Author):

The authors want to suggest a role of clock protein for risk factor to CKD induced cardiac inflammation and fibrosis in 5/6 Nx mice. In this model, they found high GPR expression in monocytes, the mechanism might be explained retinol expression in serum of CKD patients. They found the upstream binding site for clock and ANTR1.

They also showed the possibility of IL6 and TNF α overexpression for cardiac failure.

I think this MS is very interesting if they showed minor revision.

Minor comments..

You should add protein level data for Clock and Arntl in Fig 5a, because mRNA level difference is not so big.

Reviewer #3 (Remarks to the Author):

General Comments

This is a well-written and well-argued presentation of experiments examining the mechanisms by which marked reduction in renal function is linked to cardiac fibrosis and failure in a mouse model. The experiments are designed to specifically evaluate the role and presence of a direct humoral mechanism mediating macrophage-derived cardiac dysfunction by activating marrow-derived macrophages to enter into an inflammatory state (through clock protein stimulation of increased serum retinols). The experiments described narrowly but extensively investigate that hypothesis and test it. The primary strategy begins with a genetic mutation knocking out CLK, thereby disabling function of the clock gene. Utilizing a nephrectomy (5/6 Nx) model of renal dysfunction induces pathologic effects on the heart remotely. The evidence suggests that, among the responses to renal CLK protein may be a major pathological induction of increased blood retinol. Subsequent studies demonstrate that retinol induces inflammatory phenotype in circulating marrow-derived monocytes associated with the induction of GPR68 which is critical to a downstream inflammatory phenotype. Inflammatory leukocytes invade the myocardium and appear to be a major source of increased fibrosis as well as hypertrophic dysfunction. As an adjunct, a study is provided demonstrating the possibility that this might occur clinically in humans who similarly have elevated retinol levels in CKD. The manuscript presents an enormous amount of data and is well organized; nevertheless, a diagram demonstrating the proposed mechanism would help the reader follow the discussion. In addition, the proposed mechanism is restricted to

the monocyte activation:

1. While the experiments presented are very convincing with respect to the hypothesis proposed, there is little examination of the mechanism by which chemoattraction is occurring or its organ specificity. The authors assume that the invasion of circulating monocytes in this manner is sufficient for specific chemoattraction to organs. This is a unique model and so a clear mechanism by which chemoattraction occurs must be addressed. There is extensive study of inflammatory myocardial disease mechanisms that control the uptake and maturation of monocytes into macrophages. These mechanisms all include organ specificity and downstream pathological response. As an example, in the myocardial infarction primarily targeting repair, there is evidence of specific targeting by induction of leukocyte adhesion molecules in cardiac venules. In hypertrophy and fibrosis models of cardiomyopathies or elevated blood pressure arises in localized inflammation and alterations in mesenchymal cells and/or extracellular matrix. The absence of any detail of cellular mechanism target in the heart presents a picture of high levels of "killer monocytes" that attach spontaneously to the heart and induce injury with no specificity of cellular or molecular localization or further activation.
2. A similar comment can be made regarding the role of retinol and/or GPR68 made for experiments in normal mice rapidly determine whether these functions can, in themselves, directly affect the heart. The likelihood of this is improbable in the absence of the pathophysiological events.
3. Ultimately, the role of localization in the heart without mechanism presents a major problem. As presented here, one would expect the monocytes to go to other organs as well and be sensitive to similar things. It is more likely that the changes in the monocytes potentiate induction of dysregulated systemic inflammation. The cardiac response may arise from this.

Specific Comments

1. Apropos of the above, it is likely that you may have tissue in which you can examine the induction of adhesion molecules in various vascular beds in the presence and absence of retinol and at least attempt to approach that mechanism.
2. Experiments, as presented, do not examine cardiac function in any great detail but your study gives the impression that knocking out the clock protein allows the hypertrophy while angiotensin induced hypertension persists. Therefore, this mechanism was specific for altered myocardial metabolism as a result of monocyte entry independent of usual mechanistic correlation... a real conundrum.

Summary

I am impressed with the work in this paper and the dissection of an interesting mechanism by which the clock protein in a damaged organ can influence a more general inflammatory and fibrotic mechanism. The authors need to consider the organ specificity and downstream inflammatory events.

Reviewer #4 (Remarks to the Author):

The study by Yoshida et al. aimed to explore the role of dysfunction of the circadian clock on cardiac abnormalities observed in CKD mice. The authors use CKD mice model by two-step 5/6 nephrectomy, which is a valid model of progressing CKD in mice, and mutated clock gene mice to explore their hypothesis. In vitro work with circulating monocytes led them to identify the G protein-coupled receptor 68 (GPR68) as a key molecule to induce cardiac inflammation and fibrosis. Finally, they show that the increasing serum retinol levels observed in CKD mice lead to enhance the expression of GPR68 in circulating monocytes via altered CLOCK activation.

The study's rationale is original, and the used CKD mice models and in vitro experiments were adequate, however, one point should be explored:

1) Since retinoids are associated with either cellular membranes or bound to Retinol binding protein (RBP)s, it is not clear to the reviewer, whether the action of retinol is direct in the present paper. The figure 7 indicated the presence of RBP4 is necessary to the delivery of retinol to the target cells. The RBP4 serum levels are increased significantly in CKD patients, and the unbound form of RBP4 may have direct independent effects on endothelial cells. RBP4 could act as a hepatokine in the regulation of glucose metabolism by the circadian clock. Therefore, the authors need to evaluate the levels of RBP4 in their models as well as explore its role on in vitro circulating monocytes.

Reviewer #1 (Remarks to the Author):

This well-written manuscript reports the negative effect of the circadian machinery during chronic kidney disease (CKD) on cardiovascular disease. The main mechanism was shown to be induction of Gpr68. These studies appear sound and add insight into mechanisms of CKD and its effects on heart disease.

The major concerns are those sections of the manuscript which correlate retinol status to Gpr58 and other sequelae of CKD and its effects on heart disease. The connection with retinol is weak for both CKD and heart disease. It is based on literature that may be unsound, correlative, and on tenuous extrapolations by the authors. The emphasis on retinol in the Abstract and in the later figures detract from an otherwise strong presentation.

Reply to comments by reviewer#1:

We really appreciate your highly constructive comments and your opinion that “these studies appear sound and add insight into mechanisms of CKD and its effects on heart disease”. According to your suggestions, we have revised the manuscript as follows:

1. Part of the rationale for this investigation was a report (Ref. 26) correlating increased RBP4 levels (not retinol levels) with CKD inducing heart disease. Although RBP4 is a carrier for retinol, functions of RBP4 also differ from those of retinol. Moreover, the reference cited did not use an assay for RBP4 that has been established as valid (PMID17294166). In addition, the reference concluded that CKD also correlates with age, BMI, and current smoking status. The method used to measure RBP4 in reference 26 reported a median value of 34 mg/dL. This is 10-fold higher than the concentration reported in normal humans. Another article (PMID: 17618858) reported that RBP4 correlates with obesity, type II diabetes, and glucose intolerance. Therefore, the claim that retinol correlates with CKD overlooks the many other correlates. The reference (26) did not establish a sound correlation between retinol and CKD. Nor does the present manuscript do so, or establish a causative relationship.

A case in point is this statement “... the abnormal increase in serum retinol levels in 5/6Nx wild-type mice increased retinol inflammation, further exacerbating the pathologies of CKD. In contrast, 5/6Nx-induced renal inflammation was attenuated in *Clk/Clk* mice, although they had high serum retinol levels”. First, what is “retinol inflammation”? Second, doesn't this observation disconnect retinol from “retinol inflammation” from CKD caused heart disease?

We apologize for confusing you by inappropriate citation and indefinite statements. We agree with your opinion. The report (Ref.26 in the previous version) was not appropriate as the rationale for this investigation. The phrase “retinol inflammation” was also not acknowledged in general. In fact, increasing serum levels of retinol and RBP4 are often observed in CKD patients (Jing et al., *Clin. Transl. Sci.* 9, 207-215, 2016). We also demonstrated previously that serum retinol levels were increased in 5/6Nx mice (Hamamura et al., *J. Bio Chem.* 291, 4913-4927, 2016) and the increase in serum retinol levels in 5/6Nx wild-type mice induces activation of caspase and apoptotic cell death in kidney, further exacerbating the pathologies of CKD (Hamamura et al., *J. Bio Chem.* 291, 4913-4927, 2016). We also found that 5/6Nx-induced renal inflammation and apoptotic cell death were attenuated in *Clk/Clk* mice, despite exhibiting high serum levels of retinol and RBP4 (Matsunaga et al., *E bio Med.* 13, 262-273, 2016). However, it remains unknown whether elevated retinoid levels in CKD patients is related to the cardiovascular disorders. Based on these facts, we revised the 3rd paragraph of INTRODUCTION section (P.2, line 20 – P.3, line 3) to clearly inform the rationale for this investigation and presented new data as Supplementary Fig.1 showing serum levels of retinol, retinoic acid, and RBP4 in wild-type and *Clk/Clk* mice with 5/6Nx. In relation to this revision, we re-performed the experiment measuring serum levels of

retinol and retinoic acid by liquid chromatography/mass spectrometry (LC-MS/MS). In addition, RBP4 concentrations were measured by the previous method established as valid (Graham et al., *Diabetologia* 50, 814-823, 2007).

2. The statement that “Serum retinol concentrations were measured as described previously⁷²” does not seem accurate. The innovation in the reference (ref 72) was an extraction protocol that improved recovery. The current manuscript used a different extraction protocol and did not provide other essential details verifying the efficacy of the retinol analysis. This is important because the authors’ base major conclusions of the current report on an increase in retinol.

The retinol concentrations in the serum of the Jcl/ICR mice at ≥ 6 μM were much higher than in humans and other mouse strains, which are ≤ 1 to 2 μM , and lower than 1 μM in most tissues (except livers) of most mice and humans. This may be a technical problem, or reveal that this strain has unusual retinoid metabolism.

Thank you for your valuable suggestion. As described above, we re-performed the measurement of serum levels of retinol and retinoic acid by LC-MS/MS as described previously (Kane et al., *Biochem J.* 15, 363-369, 2005; Arnold et al., *J. Lipid. Res.* 53, 587-598, 2011). Using this method, we observed retinol concentrations in the serum of healthy mice and healthy human were ≤ 1 to 2 μM , and the concentrations in the serum of 5/6Nx mice and CKD patients were over 2 μM . All data showing serum levels of retinol and retinoic acid were revised according to the remeasurement method and detailed procedure was newly described in the MATERIALS AND METHODS section as “Determination of retinol and retinoic acid concentrations”.

3. Fig. 6c shows an increase in mRNA of Gpr68 in RAW264.7 cells after treatment with high concentrations of retinol. Retinol does not bind with high affinity (even at 10 μM) to nuclear receptors. There must be intermediary events between retinol and an increase in Gpr68 mRNA and STAT5 protein binding to the upstream regions of the Clock and Arntl genes. Also, no mention was made to purifying retinol. Commercial retinol often is contaminated with up to 4% retinal and some retinoic acid. Because of the high retinol concentrations used, one of these potent retinoids or other contaminant might account for the activity observed.

As your valuable suggestion, we checked the quality of retinol (Car# 95144, Sigma-Aldrich) and retinoic acid (Car# R2625, Sigma-Aldrich), which were used in this study, by LC-MS/MS analysis. The contents of retinoic acid in retinol reagent was less than 0.625 %. The contents of retinol in retinoic acid was undetectable. To clearly inform this point, the spectrum of retinol and retinoic acid reagents were shown in Supplementary Fig. 20.

We also re-performed the experiment to investigate the effects of retinol on the expression of *Gpr68* using primary culture monocytes prepared from mouse. Recent studies demonstrated that STRA6 not only functions as a retinol-bound RBP4 transporter, but also as a ligand-activated cell surface signaling receptor that, upon binding of the retinol-bound RBP4, activates JAK2/STAT5 signaling, inducing the expression of STAT target genes (Berry et al., *Proc. Nat. Acad. Sci. USA.* 108, 4340-4345, 2011). Therefore, primary culture monocytes were treated with retinol in the presence of RBP4.

The mRNA levels of *Gpr68* in mouse primary monocytes were dose-dependently increased by treatment with retinol and RBP4. Significant *Gpr68* mRNA expression was detected when cells were treated with greater than 2 μM retinol and RBP4 ($P < 0.05$), and 3 μM of retinol and RBP4 was sufficient to increase the protein levels of GPR68. In addition, treatment of mouse primary monocytes with 3 μM of retinol and RBP4 also increased the phosphorylation of nuclear STAT5 (phos-STAT5) protein, and increased the mRNA levels of *Clock* and *Arntl*. The JASPAR analysis for up- and downstream regions from the transcriptional start site revealed no possible STAT5 binding site in the mouse *Gpr68* gene, but several STAT5 binding sites were located in the

upstream regions of the *Clock* and *Arntl* genes. The chromatin immunoprecipitation experiment revealed that the amount of STAT5 binding to the upstream region of *Clock* and *Arntl* genes in mouse primary monocytes was increased by treatment with 3 μ M of retinol and RBP4. A similar increase in STAT5 binding in the upstream region of *Clock* and *Arntl* genes was observed in monocytes collected from 5/6Nx mice at 8 weeks after nephrectomy ($P < 0.01$, Fig. 6i). In addition, treatment of mouse primary monocytes with 3 μ M of retinol and RBP4 also increased CLOCK/ARNTL binding to upstream region of *Gpr68* ($P < 0.01$, Fig. 6j), indicating that retinol-induced expression of GPR68 is caused by CLOCK/ARNTL-mediated transactivation.

Downregulation of STRA6 in mouse primary monocytes by siRNA failed to induce the phosphorylation of STAT5 and the expression of CLOCK, ARNTL and GPR68, even when the cells were incubated in the media supplemented with 10% serum collected from 5/6Nx mice. After intracellular permeation, retinol is converted into retinoic acids, and then activates retinoic acid receptor (RAR) and retinoid X receptor (RXR). These receptors form heterodimer and induces the expression of their target genes. The retinol-free (unbound) RBP4 also stimulates Toll-like receptor 4 (TLR4) and activates MAP kinases. The elevation of mRNA levels for *Gpr68*, *Clock*, and *Arntl* as well as enhanced phosphorylation of STAT5 were not observed in the cultured monocytes treated with retinoic acid or RBP4 alone. Furthermore, treatment of monocytes with RAR inhibitor Ro41-5253 had a negligible effect on the expression of *Gpr68*, *Clock*, *Arntl*, and phosphorylation state of STAT5. These results indicate that STRA6 is crucial for retinol-induced expression of GPR68 in monocytes. In addition, expression of GPR68 in monocytes did not be affected by retinoic acid reagent at concentration that could be contaminated in retinol reagent. In order to clearly discuss those points, data were newly incorporated into the revised manuscript and the following points were revised.

- Data and legends were newly incorporated into Fig. 6c-6k, Supplementary Fig. 10, Supplementary Fig. 12a, and Supplementary Fig. 15 of the revised manuscript.
- The 1st paragraph of subsection "Serum retinol accumulation during chronic renal failure induces GPR68 expression in mouse monocytes" in the RESULTS section was rewritten in consideration of this result (P.8, line 23 – P.9, line 32).
- The 6th and 7th paragraphs of the DISCUSSION section were newly added in consideration of these results (P.14, line 1-19).
- Additional explanations about experimental procedure was newly added to the subsection "Cell culture and treatment (P.16, line 15-31)", "Western blotting (P.17, line 20-34)", and "Chromatin immunoprecipitation analysis (P.18, line 18-34)" in the MATERIALS AND METHODS section.

4. Fig. 6g shows an increase with high retinol treatment of STAT binding to upstream regions of *Clock* and *Arntl* genes, but in RAW264.7 cells. The essential experiment should have been done with monocytes/macrophages harvested from WT mice and humans to show a direct effect of (purified) retinol on *Gpr68* expression and STAT5 binding to the upstream regions of *Clock* and *Arntl* genes. Some experiments were done with monocytes from mice, which demonstrates the feasibility of performing the crucial experiments, while using lower concentrations of purified retinol.

As described above, we re-performed the experiment to investigate the effects of retinol on the expression of *Gpr68* STAT5 binding to the upstream regions of *Clock* and *Arntl* genes using primary culture monocytes prepared from mouse. Since there was no significant contaminant retinoic acid in our used retinol reagent, primary culture monocytes were treated with commercially available retinol in the presence of RBP4. The amount of STAT5 binding to the upstream region of *Clock* and *Arntl* genes in mouse primary monocytes was increased by treatment with 3 μ M of retinol and RBP4. Similar increase in STAT5 binding in the upstream region

of CLOCK and ARNTL genes was also observed in human primary monocytes treated with 3 μ M of retinol and RBP4. In order to clearly inform these points, data were newly incorporated into the revised manuscript and the following points were revised.

- Data and legends were newly incorporated into Fig. 6h and Fig.7d of the revised manuscript.
- The 2nd paragraph of subsection “Serum retinol accumulation during chronic renal failure induces GPR68 expression in mouse monocytes (P.8, line 23 – P.9 line 32)” and the 1st paragraph of subsection “Retinol-induced expression of GPR68 in human monocytes (P.10, line 13 - 27)” in the RESULTS section were rewritten in consideration of this result.
- Additional explanations about experimental procedure was newly added to the subsection “Cell culture and treatment (P.16, line 15-31)”, “Western blotting (P.17, line 20-34)”, and “Chromatin immunoprecipitation analysis (P.18, line 18-34)” in the MATERIALS AND METHODS section.

5. Fig. 7e, which correlates serum retinol with creatinine levels in CKD patients, has only two data points for each of the high serum levels.

According to your suggestion, we increased the number of serum samples collected from CKD patients (17 samples) and healthy subjects (8 samples). Previous report indicates that expression levels of RBP4 correlates with obesity, type II diabetes, and glucose intolerance (Klötting et al., *Cell Metab.* 6, 79-87, 2007), we collected serum from CKD patients without diagnosed diabetes. There were no abnormal levels of total protein, total cholesterol, and sugar concentrations in collected serum samples. In addition, RBP4 levels in serum of CKD patients were not correlated with the concentrations of total-, HDL- and LDL-cholesterol. In contrast, serum retinol levels were positively correlated with RBP4 levels. Both retinol and RBP4 levels in the serum of CKD patients were significantly higher than healthy subjects and those levels in CKD patients were also increased depending on serum creatinine levels. These data suggest that serum retinol and RBP4 levels increase with deteriorating renal function. In order to clearly discuss those points, data were newly incorporated into the revised manuscript and the following points were revised.

- Data and legends were newly incorporated into Fig. 7f and Supplementary Fig. 14, of the revised manuscript.
- The 2nd paragraph of subsection “Retinol-induced expression of GPR68 in human monocytes” in the RESULTS section was rewritten in consideration of this result (P.10, line 28-P.11, line 2).
- Additional explanations about experimental procedure was newly added to the subsection “Cell culture and treatment (P.16, line 15-31)” and “Western blotting (P.17, line 20-34)” in the MATERIALS AND METHODS section.

6. Fig. 7f entitled “Correlation of serum retinol levels of CKD patients with mRNA expression of GPR68, IL-6, or TNF α in THP-1 cells”, was not only a correlation, but the correlation was made with an established cell line, not with cells from CKD patients, nor in primary cells from mice.

As your suggestion, we used human primary culture monocytes to investigate the relationship between serum retinol levels in CKD patients and the expression of GPR68, IL-6, or TNF α . Incubation of human primary monocytes in the media supplemented with 20% serum of each individual CKD patient and measured the mRNA levels of *GPR68*. Induction of mRNA expression for *GPR68*, *TNF α* , and *IL-6* in human primary monocytes was detected when cells were incubated in 20% serum media. The mRNA expression levels of *GPR68*, *TNF α* , and *IL-6* in monocytes were significantly increase depending on serum retinol levels ($P < 0.01$ respectively). The production of TNF α and IL-6 from human primary monocytes was also enhanced by incubation of cells with serum of CKD patients. These data were newly incorporated into the revised manuscript and the following points were revised.

- Data and legends were newly incorporated into Fig. 7g of the revised manuscript.
- The 2nd paragraph of subsection “Retinol-induced expression of GPR68 in human monocytes” in the RESULTS section was rewritten in consideration of this result (P.11, line 2-8).
- Additional explanations about experimental procedure was newly added to the subsection “Cell culture and treatment (P.16, line 15-31)” in the MATERIALS AND METHODS section.

7. Contrary to the statement, Stra6 does not act as a receptor for retinol. It binds only RBP4. This is more than a semantic difference.

As pointed out in this comment, the function of STRA6 was rewritten as follows: “STRA6 not only functions as a retinol-bound RBP4 transporter, but also as a ligand-activated cell surface signaling receptor that, upon binding of the retinol-bound RBP4, activates JAK2/STAT5 signaling, inducing the expression of STAT target genes”.

8. The statement “Our present study also revealed the monocytic expression of Gpr68 in 5/6Nx wild-type mice activated by CLOCK/ARNTL, whose transcriptional activity was increased through the increase in serum retinol levels” is not substantiated by the data because: 1) no evidence of this was shown in primary monocytes from humans or mice; the phenomenon was induced by high retinol concentrations that were not shown to occur in CKD patients.

As suggested, we used human primary culture monocytes to investigate the relationship between serum retinol levels in CKD patients and the expression of CLOCK and ARNTL. Serum retinol levels in all CKD patients were over 2 μ M and its mean value was 3.91 μ M. Treatment of human primary monocytes treated with 3 μ M of retinol and RBP4 significantly increased STAT5 binding in the upstream region of CLOCK and ARNTL genes. The mRNA expression levels of CLOCK and ARNTL were also significantly increased depending on serum retinol levels. The published normal range of serum BNP levels is under 40 pg/mL, whereas heart failure can be diagnosed when the BNP level is over 100 pg/mL (Piotr et al., *Eur. Heart J.* 37, 2129-2200, 2016). Monocytic expression of GPR68 mRNA and serum retinol levels in CKD patients with BNP over 100 pg/mL were significantly higher than those observed in healthy subjects with BNP under 40 pg/mL ($P < 0.05$ for GPR68 mRNA, $P < 0.01$ for retinol). The levels of GPR68 mRNA and serum retinol also exhibited the increases in CKD patients with BNP between 40 and 100 pg/mL. Therefore, the serum accumulation of retinol (and also RBP4) during chronic renal failure induces GPR68 expression in monocytes. This may underlie the cardiac complications in CKD patients. These data were newly incorporated into the revised manuscript and the following points were revised.

- Data and legends were newly incorporated into Fig. 7h and 7i of the revised manuscript.
- The 2nd paragraph of subsection “Retinol-induced expression of GPR68 in human monocytes” in the RESULTS section was rewritten in consideration of this result (P.11, line 9-18).
- Additional explanations about experimental procedure was newly added to the subsection “Cell culture and treatment (P.16, line 15-31)” in the MATERIALS AND METHODS section.

9. The model in 7G is not justified by the data.

Based on the revision and newly incorporated data, we revised schematic diagram indicating the retinol/RBP4-induced expression of *Gpr68* in circulating monocytes through activation of *Clock/Arntl* and the pathological role of GPR68-expressing monocytes in the CKD-induced inflammation and fibrosis. This schematic model was presented as Fig. 8 in the revised manuscript.

Reviewer #2 (Remarks to the Author):

The authors want to suggest a role of clock protein for risk factor to CKD induced cardiac inflammation and fibrosis in 5/6Nx mice. In this model, they found high GPR expression in monocytes, the mechanism might be explained retinol expression in serum of CKD patients. They found the upstream binding site for clock and ARNTL. They also showed the possibility of IL6 and TNF α overexpression for cardiac failure. I think this MS is very interesting if they showed minor revision.

Reply to comments by reviewer#2:

We sincerely appreciate your highly evaluation for our work. According to your suggestions, we have revised the manuscript as follows:

1. You should add protein level data for Clock and Arntl in Fig 5a, because mRNA level difference is not so big.

As your suggestion, the protein levels of CLOCK and ARNTL in monocytes were assessed at ZT2 and ZT14, because the bindings of CLOCK and ARNTL on the upstream region of the mouse *Gpr68* gene was investigated at the same time points. As compared with Sham mice, the protein levels of CLOCK and ARNTL in circulating monocytes of 5/6Nx mice were increased at both time points. This result was consistent with present findings that the bindings of CLOCK and ARNTL on the upstream region of the mouse *Gpr68* gene were increased in 5/6Nx mice. These data were newly incorporated into the revised manuscript and the following points were revised.

- Data and legends were newly incorporated into Fig. 5b of the revised manuscript.
- The 1st paragraph of subsection "Disruption of the circadian rhythm of clock gene expression in 5/6Nx monocytes upregulates GPR68 expression" in the RESULTS section was rewritten in consideration of this result (P.8, line 1-5).
- Additional explanations about experimental procedure was newly added to the subsection "Western blotting (P.17, line 20-34)" in the MATERIALS AND METHODS section.

Reviewer #3 (Remarks to the Author):

This is a well-written and well-argued presentation of experiments examining the mechanisms by which marked reduction in renal function is linked to cardiac fibrosis and failure in a mouse model. The experiments are designed to specifically evaluate the role and presence of a direct humoral mechanism mediating macrophage-derived cardiac dysfunction by activating marrow-derived macrophages to enter into an inflammatory state (through clock protein simulation of increased serum retinols). The experiments described narrowly but extensively investigate that hypothesis and test it. The primary strategy begins with a genetic mutation knocking out CLK, thereby disabling function of the clock gene. Utilizing a nephrectomy (5/6 Nx) model of renal dysfunction induces pathologic effects on the heart remotely. The evidence suggests that, among the responses to renal CLK protein may be a major pathological induction of increased blood retinol. Subsequent studies demonstrate that retinol induces inflammatory phenotype in circulating marrow-derived monocytes associated with the induction of GPR68 which is critical to a downstream inflammatory phenotype. Inflammatory leukocytes invade the myocardium and appear to be a major source of increased fibrosis as well as hypertrophic dysfunction. As an adjunct, a study is provided demonstrating the possibility that this might occur clinically in humans who similarly have elevated retinol levels in CKD. The manuscript presents an enormous amount

of data and is well organized; nevertheless, a diagram demonstrating the proposed mechanism would help the reader follow the discussion. In addition, the proposed mechanism is restricted to the monocyte activation:

Reply to comments by reviewer#3:

We would like to express our gratitude for your insightful and constructive comments. Based on the revision and newly incorporated data, we revised schematic diagram indicating the retinol/RBP4-induced expression of *Gpr68* in circulating monocytes through activation of *Clock/Arntl* and the pathological role of GPR68-expressing monocytes in the CKD-induced cardiac inflammation and fibrosis. This schematic model was presented as Fig. 8 and Supplementary Fig. 10a. in the revised manuscript. We also have addressed your questions and concerns as follows:

1. While the experiments presented are very convincing with respect to the hypothesis proposed, there is little examination of the mechanism by which chemoattraction is occurring or its organ specificity. The authors assume that the invasion of circulating monocytes in this manner is sufficient for specific chemoattraction to organs. This is a unique model and so a clear mechanism by which chemoattraction occurs must be addressed. There is extensive study of inflammatory myocardial disease mechanisms that control the uptake and maturation of monocytes into macrophages. These mechanisms all include organ specificity and downstream pathological response. As an example, in the myocardial infarction primarily targeting repair, there is evidence of specific targeting by induction of leukocyte adhesion molecules in cardiac venules. In hypertrophy and fibrosis models of cardiomyopathies or elevated blood pressure arises in localized inflammation and alterations in mesenchymal cells and/or extracellular matrix. The absence of any detail of cellular mechanism target in the heart presents a picture of high levels of “killer monocytes” that attach spontaneously to the heart and induce injury with no specificity of cellular or molecular localization or further activation.

Thank you for this insightful and constructive comment. According to your suggestion, we investigated infiltration of monocytes and expression of adhesion molecules in the heart, kidney, and liver of 5/6Nx mice. Although tissue infiltration of monocytes was observed in the heart and kidney of 5/6Nx mice, no obvious migration of monocytes was detected in the liver. This organ-dependent difference in monocyte migration may be associated with distinct expression levels of adhesion molecules. The induction of VCAM1 expression in the heart and kidney of 5/6Nx mice was not affected by administration of retinol and dietary deficiency of vitamin A, suggesting that induction of adhesion molecules was not due to change in the retinol levels, but instead probably caused by elevated blood pressure and RAAS activation (Tummala et al., *Circulation* 100, 1223-1229, 1999; Riou et al., *Circ. Res.*100, 1226-1233, 2007). In contrast to the heart, GPR68-expressing cells were undetectable in the kidney of 5/6Nx mice, despite observing renal infiltration of monocytes. There are several possible reasons to explain the undetectable of GPR68-expressing monocytes in the kidney of 5/6Nx mice. Renal infiltrated GPR68-expressing monocytes in 5/6Nx mice may be eliminated by excessive activation of apoptotic signaling. Elevated serum retinol levels in 5/6Nx mice activate renal caspase activity, leading to apoptotic cell death (Hamamura et al., *J. Bio Chem.* 291, 4913-4927, 2016). On the other hand, TGF β 1 produced from injured kidney may induce differentiation of infiltrated monocytes. The expression and production of TGF β 1 were increased in the kidney, but not in the heart, of 5/6Nx mice. The expression of *Gpr68* mRNA in primary cultured monocytes was increased by incubation of cells in the media containing 10% serum collected from 5/6Nx mice, but 5/6Nx serum-induced expression of *Gpr68* mRNA was prevented by TGF β . Treatment with TGF β also induces the differentiation of Ly6C⁺ monocytes into Ly6C⁻ cells. Therefore, renal infiltrated GPR68-expressing monocytes may be differentiated into different subtypes of macrophages. Taken together, cardiac

migration of GPR68-expressing monocytes in 5/6Nx mice seemed to be associated with organ-dependent difference in the expression of adhesion molecules and tissue-specific differentiation system. To clearly inform these points, data were newly incorporated into the revised manuscript and the following points were revised. To clearly inform these points, data were newly incorporated into the revised manuscript and the following points were revised.

- Data and legends were revised and newly incorporated into Fig. 2a-2f, Fig.3e-3f, Fig.4b, Supplementary Fig.5, Supplementary Fig.6c-6d, Supplementary Fig.7, Supplementary Fig. 12d-12e, Supplementary Fig.16, Supplementary Fig.17, Supplementary Fig.18, Supplementary Fig.19 of the revised manuscript.
- The 2nd paragraph of subsection “GPR68 expression in monocyte-derived cardiac macrophages” (P.5, line 12-20), “Migration of high-GPR68-expressing monocytes into the heart during chronic renal failure (P.6, line 12-32)”, and “Downregulation of GPR68 in monocytes alleviates CKD-induced cardiac inflammation (P.6, line 34 - P.7, line 29)” were rewritten in consideration of this result.
- The 8th paragraphs of the DISCUSSION section were newly added in consideration of these results (P.14, line 8 - P.15, line 7).
- Additional explanations about experimental procedure was newly added to the subsection “Animals and treatment”, “Immunofluorescence histochemical staining” and “Microarray gene expression analysis” in the MATERIALS AND METHODS section.

2. A similar comment can be made regarding the role of retinol and/or GPR68 made for experiments in normal mice rapidly determine whether these functions can, in themselves, directly affect the heart. The likelihood of this is improbable in the absence of the pathophysiologic events.

According to your valuable suggestion, we investigated the effect of retinol on the expression of *Gpr68* in monocytes and on cardiac function of healthy mice. Vitamin A is derived from the diet mostly as retinol esters. As retinol is the main circulating retinoid, dietary deficiency of vitamin A decreases serum retinol levels (Blomhoff et al., *J Neurobiol.* 66, 606-630, 2006; Weber et al., *Mol. Nutr. Food Res.* 56, 251-258, 2012). Feeding of 5/6Nx mice with vitamin A-free diet decreased the serum levels of both retinol and RBP4, accompanied by suppressing the mRNA expressions of *Gpr68*, *Clock*, and *Arntl* in circulating monocytes. Although serum urea nitrogen levels and cardiac expression of adhesion molecule VCAM1 were not decreased in 5/6Nx mice fed with vitamin A-free diet, the deficient diet suppressed 5/6Nx-induced cardiac migration of GPR68-expressing monocytes, elevation of serum BNP levels, and fibrosis in the heart. On the other hand, intraperitoneal administration of retinol to healthy mice induced GPR68 expression in the circulating monocytes, but retinol-treated healthy mice did not show cardiac migration of monocytes, VCAM1 expression, characteristics of heart failure. These findings suggest that the CKD-induced elevation of serum retinol and RBP4 induces the monocytic expression of GPR68, and cardiac migration of GPR68-expressing cells appears to exacerbate CKD-induced heart failure. To clearly inform these points, data were newly incorporated into the revised manuscript and the following points were revised.

- Data and legends were revised and newly incorporated into Fig. 6l-6p, Supplementary Fig.11, and Supplementary Fig.12 of the revised manuscript.
- The 2nd paragraph of subsection “Serum retinol accumulation during chronic renal failure induces GPR68 expression in mouse monocytes” was revised in consideration of this result (P.9, line 33 - P.10, line 10).
- Additional explanations about experimental procedure was newly added to the subsection “Animals and treatment” in the MATERIALS AND METHODS section.

3. Ultimately, the role of localization in the heart without mechanism presents a major problem.

As presented here, one would expect the monocytes to go to other organs as well and be sensitive to similar things. It is more likely that the changes in the monocytes potentiate induction of dysregulated systemic inflammation. The cardiac response may arise from this. Apropos of the above, it is likely that you may have tissue in which you can examine the induction of adhesion molecules in various vascular beds in the presence and absence of retinol and at least attempt to approach that mechanism.

As described in above comments, cardiac migration of GPR68-expressing monocytes in 5/6Nx mice seemed to be associated with organ-dependent difference in the expression of adhesion molecules and tissue-specific differentiation system. Results of microarray analysis revealed that the expression levels of adhesion molecules and chemokines, involving in the regulation of monocyte infiltration, were elevated in the heart and kidney of 5/6Nx mice, but such elevations of adhesion molecules and chemokines were not detected in the liver of 5/6 mice. In fact, infiltration of monocytes was observed in the heart and kidney, but not in the liver, of 5/6Nx mice. Furthermore, the expression of adhesion molecule VCAM1 in the heart and kidney of 5/6Nx mice was not affected by administration of retinol and dietary deficiency of vitamin A, suggesting that induction of adhesion molecules was not due to change in the retinol levels, but instead probably caused by elevated blood pressure and RAAS activation (Tummala et al., *Circulation* 100, 1223-1229, 1999; Riou et al., *Circ. Res.* 100, 1226-1233, 2007). These data were newly incorporated into the revised manuscript as Supplementary Fig. 5, Supplementary Fig. 11c-11d, Supplementary Fig. 12d-12e, Supplementary Fig. 16c-16d, and Supplementary Fig. 17.

4. Experiments, as presented, do not examine cardiac function in any great detail but your study gives the impression that knocking out the clock protein allows the hypertrophy while angiotensin induced hypertension persists. Therefore, this mechanism was specific for altered myocardial metabolism as a result of monocyte entry independent of usual mechanistic correlation... a real conundrum.

As you pointed out, 5/6Nx mice exhibited increased serum BNP levels and cardiac fibrosis without reduced ejection fraction. These symptoms are similar to heart failure with preserved ejection fraction (HFpEF). The hallmark of HFpEF is left atrial dilation, often accompanied by left ventricular hypertrophy and increased stiffness (Michel et al., *Eur. Heart J.* 35, 1022-1032, 2014). Recent studies also indicate the implication of cardiac inflammation and fibrosis in the exacerbation of HFpEF (Paulus et al., *J Am. Coll. Cardiol.* 62, 263-271, 2013; Gavin et al., *J Am. Coll. Cardiol.* 70, 2186-2200, 2017). We observed monocytic expression of GPR68 and cardiac fibrosis in mice within 8 or 14 weeks after 5/6Nx operation due to the decrease in their survival rate associated with uremia, decline of glomerular filtration rate, and elevated urea nitrogen levels (Oliver et al., *PLoS One* 5, e11979, 2010; Erin et al., *J Am. Soc. Nephrol.* 21, 1678-1690, 2010). More prolonged observation of cardiac functions of 5/6Nx mice may make it possible to clarify the role of monocytic clock genes in the pathogenesis of CKD-induced heart failure. To clearly inform these points, the 2nd paragraph of DISCUSSION section was rewritten (P.12, line 3-23).

5. I am impressed with the work in this paper and the dissection of an interesting mechanism by which the clock protein in a damaged organ can influence a more general inflammatory and fibrotic mechanism. The authors need to consider the organ specificity and downstream inflammatory events.

We sincerely appreciate your highly evaluation for our work. We conducted additional experiments to understand the organ specificity of monocyte migration and their downstream inflammatory events as much as possible. Your constructive comments were very useful for improvement of the manuscript.

Reviewer #4 (Remarks to the Author):

The study by Yoshida et al. aimed to explore the role of dysfunction of the circadian clock on cardiac abnormalities observed in CKD mice. The authors use CKD mice model by two step 5/6 nephrectomy, which is a valid model of progressing CKD in mice, and mutated clock gene mice to explore their hypothesis. In vitro work with circulating monocytes led them to identify the G protein-coupled receptor 68 (GPR68) as a key molecule to induce cardiac inflammation and fibrosis. Finally, they show that the increasing serum retinol levels observed in CKD mice lead to enhance the expression of GPR68 in circulating monocytes via altered CLOCK activation. The study's rationale is original, and the used CKD mice models and in vitro experiments were adequate, however, one point should be explored:

Reply to comments by reviewer#4:

We are grateful for insightful and constructive comments. We have addressed your concern as follows:

1. Since retinoids are associated with either cellular membranes or bound to Retinol binding protein (RBP)s, it is not clear to the reviewer, whether the action of retinol is direct in the present paper. The figure 7 indicated the presence of RBP4 is necessary to the delivery of retinol to the target cells. The RBP4 serum levels are increased significantly in CKD patients, and the unbound form of RBP4 may have direct independent effects on endothelial cells. RBP4 could act as a hepatokine in the regulation of glucose metabolism by the circadian clock. Therefore, the authors need to evaluate the levels of RBP4 in their models as well as explore its role on in vitro circulating monocytes.

As pointed out in this comment, RBP4 is necessary to the delivery of retinol to the target cells. STRA6 not only functions as a retinol-bound RBP4 transporter, but also as a ligand-activated cell surface signaling receptor that, upon binding of the retinol-bound RBP4, activates JAK2/STAT5 signaling, inducing the expression of STAT target genes. Retinol is derived from dietary intake of both retinol and pro-vitamin A (Blomhoff et al., *J Neurobiol.* 66, 606-630, 2006; Weber et al., *Nutr. Food Res.* 56, 251-258, 2012), while RBP4 is mainly synthesized in the liver and secreted into the circulation (Blomhoff et al., *J Neurobiol.* 66, 606-630, 2006). Kidney is the main site of RBP4 catabolism and contributes to an elevation of RBP4 levels during chronic renal failure (Jing et al., *Clin. Transl. Sci.* 9, 207-215, 2016; Goodman et al., *Ann. NY. Acad. Sci.* 90, 378-390, 1980). Hepatic secretion of RBP4 is also enhanced by retinol (Goodman et al., *Ann. NY. Acad. Sci.* 90, 378-390, 1980; Ronne et al., *J Cell Biol.* 96, 907-910, 1983). In fact, administration of retinol to normal mice increased serum RBP4 levels.

Elevated serum levels of retinol and RBP4 were observed in 5/6Nx mice. Furthermore, both retinol and RBP4 levels in the serum of CKD patients were also significantly higher than those in healthy subjects. There was a correlation between retinol and RBP4 levels in the human serum samples. Therefore, we re-performed experiments to investigate the effect of retinol on the mouse and human primary monocytes in the presence of RBP4. As expression levels of GPR68 and inflammation cytokines in the mouse and human primary monocytes were increased depending on retinol/RBP4 levels, elevation of serum levels of retinol and RBP4 could be an inducible factor for CKD-induced heart failure. These data were newly incorporated into the revised manuscript and the following points were revised.

- Data and legends were revised and newly incorporated into Fig. 6b-6k, Fig.7a-7i, Supplementary Fig.10, Supplementary Fig.12, and Supplementary Fig.15 of the revised manuscript.
- The 1st paragraph of subsection "Serum retinol accumulation during chronic renal failure

induces GPR68 expression in mouse monocytes (P.8, line 23 - P.9, line 32)” and “Retinol-induced expression of GPR68 in human monocytes (P.10, line 12 - P.11, line 18)” in the RESULTS section were rewritten in consideration of this result.

- The 6th and 7th paragraphs of the DISCUSSION section were newly added in consideration of these results (P.14, line 1 - 19).
- Additional explanations about experimental procedure was newly added to the subsection “Cell culture and treatment (P.16, line 15 - 31)” and “Western blotting (P.17, line 20 - 34)” in the MATERIALS AND METHODS section.

We also re-checked and amended the references throughout the text. The reference section was reorganized according to the revision of manuscript.

REVIEWER COMMENTS

Reviewer #3 (Remarks to the Author):

This reviewer is very impressed with the authors devotion to the mechanisms they have described and the great effort they have made to comply with four rather complex reviews. I am reviewer #3 and the experiments presented to better understand cardiac specific inflammatory reactions in this model were done very well. It is Impossible to provide a criticism-proof response to some questions originally raised, but the basic concerns with regard to the importance of these mechanisms has clearly been allayed. Moreover, some of the new experiments provide interesting directions in which the author might pursue their muse.

MARK L. ENTMAN, MD

Reviewer #4 (Remarks to the Author):

The authors did answer as they can adequately to my requests
I have nothing to add

Reviewer #5 (Remarks to the Author):

Dear authors,

You have made an extensive effort to accommodate the comments that you received. Still, a few concerns remain.

You state several times, that the model used is clinically relevant. I think this is an overstatement as it does not resemble the progressive loss of nephrons in glomerular CKD's and you can discuss whether it is a chronic model also. Would it not be more correct to say that it is a frequently used model?

You conclude (title, abstract etc) that Gpr68 exacerbates cardiac disease. You find an almost normalization of cardiac measurements in fig 1, g, h and I, when you suppress Gpr68 expression by siRNAs and when you suppress Gpr68 in Splx5/6Nx mice (figure 4). I am amazed that you can prevent cardiac disease by the interventions, but it is what your data shows. Thus, would it not be more correct to write that you can prevent cardiac inflammation by Gpr68 inhibition?

Please note several errors in figure legend to supplementary figure 12 (etinol => retinol, numbering).

Please, see a few more comments below.
Thanks.

1. Part of the rationale for this investigation was a report (Ref. 26) correlating increased RBP4 levels (not retinol levels) with CKD inducing heart disease. Although RBP4 is a carrier for retinol, functions of RBP4 also differ from those of retinol. Moreover, the reference cited did not use an assay for RBP4 that has been established as valid (PMID17294166). In addition, the reference concluded that CKD also correlates with age, BMI, and current smoking status. The method used to measure RBP4 in reference 26 reported a median value of 34 mg/dL. This is 10-fold higher than the concentration reported in normal humans. Another article (PMID: 17618858) reported that RBP4 correlates with obesity, type II diabetes, and glucose intolerance. Therefore, the claim that retinol correlates with CKD overlooks the many other correlates. The reference (26) did not establish a sound correlation between retinol and CKD. Nor does the present manuscript do so, or establish a causative relationship.

A case in point is this statement "... the abnormal increase in serum retinol levels in 5/6Nx wildtype

mice increased retinol inflammation, further exacerbating the pathologies of CKD. In contrast, 5/6Nx-induced renal inflammation was attenuated in Clk/Clk mice, although they had high serum retinol levels". First, what is "retinol inflammation"? Second, doesn't this observation disconnect retinol from "retinol inflammation" from CKD caused heart disease?

We apologize for confusing you by inappropriate citation and indefinite statements. We agree with your opinion. The report (Ref.26 in the previous version) was not appropriate as the rationale for this investigation. The phrase "retinol inflammation" was also not acknowledged in general. In fact, increasing serum levels of retinol and RBP4 are often observed in CKD patients (Jing et al., Clin. Transl. Sci. 9, 207-215, 2016). We also demonstrated previously that serum retinol levels were increased in 5/6Nx mice (Hamamura et al., J. Bio Chem. 291, 4913-4927, 2016) and the increase in serum retinol levels in 5/6Nx wild-type mice induces activation of caspase and apoptotic cell death in kidney, further exacerbating the pathologies of CKD (Hamamura et al., J. Bio Chem. 291, 4913-4927, 2016).

Just a comment here; why is kidney improvement not found in the Vitamin A deficient experiment in the present study? As far as I can see this is not the case.

You further write (Page 2, line 35 and 36): "In contrast, 5/6Nx-induced renal inflammation and apoptotic cell death were attenuated in Clk/Clk mice even though they exhibited high serum levels of retinol and RBP4 (Supplementary Fig. 1)".

But again, why do you not see kidney improvement during retinol deficiency in this study, if you think retinol induced CLOCK genes are involved in renal disease? And where is it shown that Clk/Clk mice have attenuation of renal inflammation and apoptotic cell death? Not in Supplementary fig 1. Please, give correct reference.

We also found that 5/6Nx-induced renal inflammation and apoptotic cell death were attenuated in Clk/Clk mice, despite exhibiting high serum levels of retinol and RBP4 (Matsunaga et al., E bio Med. 13, 262-273, 2016). However, it remains unknown whether elevated retinoid levels in CKD patients is related to the cardiovascular disorders.

In the changed Introduction on page 2 line 35 you make reference to 24 (Matsunaga et al.): "In characterizing the hepatic metabolic status and inflammatory response of wild-mice with 5/6 nephrectomy (5/6Nx), a clinically relevant model of CKD, we found that the abnormal increase in serum retinol levels in 5/6Nx wild-type mice induces the activation of caspase and apoptotic cell death in the kidney, further exacerbating the pathologies of CKD 24."; Is this the correct reference?

Based on these facts, we revised the 3rd paragraph of INTRODUCTION section (P.2, line 20 – P.3, line 3) to clearly inform the rationale for this investigation and presented new data as Supplementary Fig.1 showing serum levels of retinol, retinoic acid, and RBP4 in wild-type and Clk/Clk mice with 5/6Nx. In relation to this revision, we re-performed the experiment measuring serum levels of retinol and retinoic acid by liquid chromatography/mass spectrometry (LC-MS/MS). In addition, RBP4 concentrations were measured by the previous method established as valid (Graham et al., Diabetologia 50, 814-823, 2007).

Still, you terminate the section (page 2, line 1-3) with the sentence: "...it remains unknown whether high retinoid levels in CKD patients are related to cardiovascular disorder";. But this is not your primary focus in this paper and it is not reflected in the title of your paper either. So, what is the aim of the present study? Would it not be more correct to say that you wish to investigate cardiac outcome in the Clk/Clk model with kidney impairment?

9. The model in 7G is not justified by the data.

Based on the revision and newly incorporated data, we revised schematic diagram indicating the retinol/RBP4-induced expression of Gpr68 in circulating monocytes through activation of Clock/Arntl and the pathological role of GPR68-expressing monocytes in the CKD-induced inflammation and fibrosis. This schematic model was presented as Fig. 8 in the revised manuscript.

It should be emphasized that the Gpr68 expression in monocytes does not per se initiate migration of monocytes to the cardiac wall if renal disease is not present.

Reviewer #5 (Remarks to the Author):

Dear authors,

You have made an extensive effort to accommodate the comments that you received. Still, a few concerns remain.

Reply to comments by reviewer#5 (related to reviewer#1's comments):

We are grateful for insightful and constructive comments. We have addressed your concern as follows:

You state several times, that the model used is clinically relevant. I think this is an overstatement as it does not resemble the progressive loss of nephrons in glomerular CKD's and you can discuss whether it is a chronic model also. Would it not be more correct to say that it is a frequently used model?

According to your suggestion, we revised the phrases of 5/6Nx mice as follows (P2. line 32- 35, P3. line 25- 26): "mice with 5/6 nephrectomy (5/6Nx), an experimental model of CKD characterized by the slow development of glomerulosclerosis, vascular sclerosis, tubulointerstitial fibrosis and renal inflammation" and "Subtotal nephrectomy, termed 5/6Nx, induces the progressive renal failure involving glomerulosclerosis and tubulointerstitial fibrosis".

You conclude (title, abstract etc) that Gpr68 exacerbates cardiac disease. You find an almost normalization of cardiac measurements in fig 1, g, h and I, when you suppress Gpr68 expression by siRNAs and when you suppress Gpr68 in Splx5/6Nx mice (figure 4). I am amazed that you can prevent cardiac disease by the interventions, but it is what your data shows. Thus, would it not be more correct to write that you can prevent cardiac inflammation by Gpr68 inhibition?

As your suggestion, we changed the title as follows "Alteration of circadian machinery in monocytes underlies the exacerbation of cardiac inflammation and fibrosis associated with chronic kidney disease". We also re-wrote throughout the manuscript to clearly inform that inhibition of monocytic Gpr68 expression can prevent cardiac inflammation and fibrosis.

Please note several errors in figure legend to supplementary figure 12 (etinol => retinol, numbering).

We corrected the errors in figure legend to supplementary figure 12.

1. Part of the rationale for this investigation was a report (Ref. 26) correlating increased RBP4 levels (not retinol levels) with CKD inducing heart disease. Although RBP4 is a carrier for retinol, functions of RBP4 also differ from those of retinol. Moreover, the reference cited did not use an assay for RBP4 that has been established as valid (PMID17294166). In addition, the reference concluded that CKD also correlates with age, BMI, and current smoking status. The method used to measure RBP4 in reference 26 reported a median value of 34 mg/dL. This is 10-fold higher than the concentration reported in normal humans. Another article (PMID: 17618858) reported that RBP4 correlates with obesity, type II diabetes, and glucose intolerance. Therefore, the claim that retinol correlates with CKD overlooks the many other correlates. The reference (26) did not establish a sound correlation between retinol and CKD. Nor does the present manuscript do so, or establish a causative relationship.

A case in point is this statement "... the abnormal increase in serum retinol levels in 5/6Nx wildtype mice increased retinol inflammation, further exacerbating the pathologies of CKD. In contrast,

5/6Nx-induced renal inflammation was attenuated in Clk/Clk mice, although they had high serum retinol levels". First, what is "retinol inflammation"? Second, doesn't this observation disconnect retinol from "retinol inflammation" from CKD caused heart disease?

We apologize for confusing you by inappropriate citation and indefinite statements. We agree with your opinion. The report (Ref.26 in the previous version) was not appropriate as the rationale for this investigation. The phrase "retinol inflammation" was also not acknowledged in general. In fact, increasing serum levels of retinol and RBP4 are often observed in CKD patients (Jing et al., Clin. Transl. Sci. 9, 207-215, 2016). We also demonstrated previously that serum retinol levels were increased in 5/6Nx mice (Hamamura et al., J. Bio Chem. 291, 4913-4927, 2016) and the increase in serum retinol levels in 5/6Nx wild-type mice induces activation of caspase and apoptotic cell death in kidney, further exacerbating the pathologies of CKD (Hamamura et al., J. Bio Chem. 291, 4913-4927, 2016).

Just a comment here; why is kidney improvement not found in the Vitamin A deficient experiment in the present study? As far as I can see this is not the case. You further write (Page 2, line 35 and 36): "In contrast, 5/6Nx-induced renal inflammation and apoptotic cell death were attenuated in Clk/Clk mice even though they exhibited high serum levels of retinol and RBP4 (Supplementary Fig. 1)". But again, why do you not see kidney improvement during retinol deficiency in this study, if you think retinol induced CLOCK genes are involved in renal disease?

As indicated in Supplementary Fig. 11a, feeding of 5/6Nx mice with a vitamin A-free diet, that was initiated at 8 weeks after nephrectomy, prevented the increase in serum BNP levels and cardiac fibrosis without suppression of renal dysfunction. On the other hand, our previous study demonstrated that initiation of dietary vitamin A-deficiency in 5/6Nx mice at 4 weeks after nephrectomy suppressed renal dysfunction with elevation of serum urea nitrogen levels²⁶. This discrepancy may be due to the difference in the degree of renal impairment at the initiation of vitamin A-deficiency. Serum levels of urea nitrogen and creatinine in 5/6Nx mice begun to increase around 6 weeks after nephrectomy. Renal function of 5/6Nx mice might have been irreversibly impaired at 8 weeks after nephrectomy. Therefore, the initiation of vitamin A-deficiency in 5/6Nx mice after the exacerbation of renal dysfunction would be late to suppress the increase in serum urea nitrogen levels. In contrast to kidney, heart failure of 5/6Nx mice might be still reversible at 8 weeks after surgical nephrectomy. The inflammation response in the kidney of 5/6Nx mice is probably caused by the activation of RAR/RXR-mediated pathway, but their cardiac inflammation is dependent on the infiltration of GPR68-expressing monocytes. The mechanistic difference in the inflammation response between kidney and heart may contribute to the progression rate of their impairment. To clearly inform these points, data were newly incorporated into the revised manuscript and the following points were revised.

- Data and legends were newly incorporated into Supplementary Fig.1 of the revised manuscript.
- The 8th paragraph of DISCUSSION section was newly added in consideration of these results (P.14, line 21 - P.15, line 2).

And where is it shown that Clk/Clk mice have attenuation of renal inflammation and apoptotic cell death? Not in Supplementary fig 1. Please, give correct reference.

We apologize for confusing you by inappropriate statements. We previously reported the attenuation of renal inflammation and apoptotic cell death in Clk/Clk mice with 5/6Nx (Matsunaga et al., E bio Med. 13, 262-273, 2016). The sentence in the 3rd paragraph of INTRODUCTION section was re-written by correcting reference (P.3, line 1-2)

We also found that 5/6Nx-induced renal inflammation and apoptotic cell death were attenuated in Clk/Clk mice, despite exhibiting high serum levels of retinol and RBP4 (Matsunaga et al., *Ebio Med.* 13, 262-273, 2016). However, it remains unknown whether elevated retinoid levels in CKD patients is related to the cardiovascular disorders.

In the changed Introduction on page 2 line 35 you make reference to 24 (Matsunaga et al.): "In characterizing the hepatic metabolic status and inflammatory response of wild-mice with 5/6 nephrectomy (5/6Nx), a clinically relevant model of CKD, we found that the abnormal increase in serum retinol levels in 5/6Nx wild-type mice induces the activation of caspase and apoptotic cell death in the kidney, further exacerbating the pathologies of CKD 24."; Is this the correct reference?

We apologize for confusing you by inappropriate citation. The correct reference is as follows: Hamamura et al., *J. Biol. Chem.* 291, 4913-4927, 2016. In relation to this correction, reference section was also re-organized.

Based on these facts, we revised the 3rd paragraph of INTRODUCTION section (P.2, line 20 – P.3, line 3) to clearly inform the rationale for this investigation and presented new data as Supplementary Fig.1 showing serum levels of retinol, retinoic acid, and RBP4 in wild-type and Clk/Clk mice with 5/6Nx. In relation to this revision, we re-performed the experiment measuring serum levels of retinol and retinoic acid by liquid chromatography/mass spectrometry (LC-MS/MS). In addition, RBP4 concentrations were measured by the previous method established as valid (Graham et al., *Diabetologia* 50, 814-823, 2007).

Still, you terminate the section (page 2, line 1-3) with the sentence: ".it remains unknown whether high retinoid levels in CKD patients are related to cardiovascular disorder";. But this is not your primary focus in this paper and it is not reflected in the title of your paper either. So, what is the aim of the present study? Would it not be more correct to say that you wish to investigate cardiac outcome in the Clk/Clk model with kidney impairment?

As your suggestion, the sentence in the 3rd paragraph of INTRODUCTION section was re-written as follows: "it remains unknown whether CKD-induced increase in retinol levels exacerbates cardiovascular disorders via alteration of *Clock* gene expression" (P.3, line 4-5).

9. The model in 7G is not justified by the data.

Based on the revision and newly incorporated data, we revised schematic diagram indicating the retinol/RBP4-induced expression of Gpr68 in circulating monocytes through activation of Clock/Arntl and the pathological role of GPR68-expressing monocytes in the CKD-induced inflammation and fibrosis. This schematic model was presented as Fig. 8 in the revised manuscript.

It should be emphasized that the Gpr68 expression in monocytes does not per se initiate migration of monocytes to the cardiac wall if renal disease is not present.

As pointed out in this comment, schematic diagram shown in Fig.8 was revised to highlight the fact that CKD-associated cardiac disease is multifactorial.

REVIEWERS' COMMENTS

Reviewer #5 (Remarks to the Author):

Dear Authors

Thank you for replying to my questions.

Reviewer #5 (Remarks to the Author):

Thank you for replying to my questions.

Reply to comments by reviewer#1:

We really appreciate your highly constructive comments and your suggestions were very useful for improving our manuscript.